# 3D cell segregation geometry and dynamics are governed by tissue surface tension regulation

Elod Méhes [1], Enys Mones[1], Máté Varga [2], Áron Zsigmond [2], Beáta Biri-Kovács[3], László Nyitray[3], Vanessa Barone[4,5], Gabriel Krens[5], Carl-Philipp Heisenberg [5] & Tamás Vicsek [1]✉

Tissue morphogenesis and patterning during development involve the segregation of cell types. Segregation is driven by differential tissue surface tensions generated by cell types through controlling cell-cell contact formation by regulating adhesion and actomyosin contractility-based cellular cortical tensions. We use vertebrate tissue cell types and zebrafish germ layer progenitors as in vitro models of 3-dimensional heterotypic segregation and developed a quantitative analysis of their dynamics based on 3D time-lapse microscopy. We show that general inhibition of actomyosin contractility by the Rho kinase inhibitor Y27632 delays segregation. Cell type-specific inhibition of non-muscle myosin2 activity by overexpression of myosin assembly inhibitor S100A4 reduces tissue surface tension, manifested in decreased compaction during aggregation and inverted geometry observed during segregation. The same is observed when we express a constitutively active Rho kinase isoform to ubiquitously keep actomyosin contractility high at cell-cell and cell-medium interfaces and thus overriding the interface-specific regulation of cortical tensions. Tissue surface tension regulation can become an effective tool in tissue engineering.

[1] Department of Biological Physics, ELTE Eötvös University, Budapest, Hungary. [2] Department of Genetics, ELTE Eötvös University, Budapest, Hungary. [3] Department of Biochemistry, ELTE Eötvös University, Budapest, Hungary. [4] Center for Marine Biotechnology and Biomedicine, University of California San Diego, La Jolla, CA, USA. [5] Institute of Science and Technology Austria, Klosterneuburg, Austria. ✉email: vicsek@hal.elte.hu

Tissue self-organization, such as the segregation of cell types based on their biomechanical properties is an important component of embryonic development in metazoans[1–3]. Well-characterized examples include the chick limb bud development[4,5], mouse blastocyst formation[6–8] as well as gastrulation and germ layer formation in the zebrafish and *Xenopus* embryos[9–12]. The emerging fields of tissue engineering and biofabrication[13] can also exploit the (uncovered) mechanisms of self-organization.

On a long timescale, tissues behave like viscous fluids characterized by specific tissue surface tension (TST), as the manifestation of cohesion, which is determined by cell adhesion and cellular cortical tension. The relative contributions of adhesion and cortical tension to TST are approached by two hypotheses: the differential adhesion hypothesis (DAH)[4,14,15] and the differential interfacial tension hypothesis (DITH)[16]. Specific differences in TST are considered as major contributors to in vitro and in vivo cell segregation/sorting and tissue layering in development, although other mechanisms like collective cell migration[17,18] and polarization[10] or osmolarity[19] clearly play crucial roles.

In the course of in vitro segregation of different cell types, the cell type characterized by higher TST tends to segregate inside, enveloped, or engulfed by cells with lower TST[4,12,14,20–22]. For generating high TST, the cell-medium interfacial tension, made up of cell cortical tension only, has to be increased whereas cell-cell interfacial tension has to be decreased, as TST depends on the ratio of cell-medium interface vs. cell-cell interface tensions. Cell-cell interfacial tension is mainly composed of cortical tension generated by actomyosin contraction while the (negative) contribution of adhesion tension is low; therefore reducing cell-cell interfacial tension requires active reduction of local cortical tension[23].

Depletion of non-muscle myosin 2 (NM2) was observed at cell-cell interfaces concomitant with apparent NM2 and actin accumulation at cell-medium interface in germ layer progenitors in vitro and in vivo[21,23]. Cortical mechanical tension was shown to promote cortical NM2 localization in *Drosophila* embryos with NM2 itself acting as a mechanosensor in this recruitment process[24,25]. The interface-specific differential regulation of cortical tension is a crucial component of TST generation and it is thought to be directed by signaling from the cell-cell adhesion complexes to the cytoskeleton[26]. This signaling pathway starts from cadherin adhesion molecules that are trans-bound with cadherins of another contacting cell and recruit intracellular catenins to form a complex. Among others, p120-catenin (catenin-delta1) is recruited and activated here, whereby it inhibits RhoA, leading to the inactivation of Rho kinase and further downstream inhibition of actomyosin contractility[27–31].

All components of the cytoskeleton contribute to cells' biomechanical properties but the actomyosin cytoskeleton is of outstanding importance. While the cortical actin network is indispensable for the generation of cortical tension[32], the main actors in the process are the myosin motor proteins. The molecular interactions unveiled in different experimental systems underline the importance of local regulation of myosin activity in the development of biomechanical characteristics required for tissue self-organization.

Motivated by earlier studies on the role of actomyosin cytoskeleton in the regulation of tissue surface tension and segregation[21,23], we extended these studies to a wider range of vertebrate cell types and developed a quantitative analysis of the dynamics of segregation.

In this paper we use different vertebrate tissue cell types and zebrafish germ layer progenitor cells as in vitro models of 3-dimensional heterotypic segregation to study how perturbations of the actomyosin contractile system influence tissue surface tension generation, the dynamics of aggregation and segregation as well as the geometry of segregated homotypic domains. We show that general pharmacological inhibition of actomyosin contractility by the Rho kinase (ROCK) inhibitor Y27632 slows down the segregation of all tested cell types without affecting the basic spatial configuration of segregated domains. We also show that cell type specific inhibition of non-muscle myosin 2 activity by overexpressing the myosin assembly inhibitor S100A4 protein decreases tissue surface tension, manifested in defective aggregation and inverted segregation positioning. By expressing constitutively active ROCK in zebrafish ectoderm cells to constantly activate actomyosin contractility at both cell-cell and cell-medium interfaces we demonstrate that overriding the interface-specific regulation of cortical tension and thereby preventing local reduction of cell-cell interface tension will result in defective contact formation. Additionally, such overriding leads to decreased tissue surface tension, manifested in reduced compaction and inverted segregation configuration, which underlines the role of ROCK in interface-specific downregulation of cortical tension for effective cell contact formation.

## Results

**Characterization of homotypic contact formation**. Cell contact formation is the basic process underlying multicellular aggregation and segregation. To characterize homotypic contact formation and adhesion we used the dual pipette aspiration (DPA) assay to measure the strength of mechanical coupling in contact forming cells. We mechanically separated homotypic cell doublets formed in cell suspensions of two model cell types: primary goldfish keratinocytes (PFK) or EPC fish keratinocyte cell line, and measured the corresponding separation forces. Homotypic doublet formation was limited to 1 min contact time and subsequently rapid ($t < 1\,s$) cell separation was achieved leaving little time for any reorganization of cell-cell contact, therefore the measurement is best described as the separation of two elastic solids. We found that homotypic doublets of PFK cells were characterized by much higher contact strength manifested in higher separation forces ($F_s$) than EPC doublets (Fig. 1a). Contact shapes were also different as PFK doublets were observed to have larger contact angles (45.1º, SEM = 0.71) than EPC doublets (35.5º, SEM = 0.65), in harmony with the differences in contact strength (Fig. 1b, c, Supplementary Movies 1 and 2).

**3D segregation of homotypic cell domains**. Segregation (or sorting) of two cell types and formation of 3D homotypic cell domains out of an even mixture of approximately equal number of cells is a self-organized process mainly controlled by cells' contact formation and cohesion, and the resultant tissue surface tension characterizing the given cell types. To observe the basic process of segregation and the final spatial positioning of segregated domains we selected three cell type pairs, mainly on the basis of earlier data on their homotypic adhesion contact strengths. First, primary goldfish keratinocytes (PFK) and EPC fish keratinocytes, characterized by markedly different homotypic contact strengths (Fig. 1a), were mixed in suspensions in non-adherent aggregation chamber wells where only cell-cell adhesion and no cell-substrate adhesion occur. After mixing, we observed the process of segregation by time-lapse fluorescent videomicroscopy and recorded the image series of individual aggregates ($n > 30$) for several hours until final spatial configurations of the segregated domains were reached. In all experiments, PFK cells were segregated inside the emerging spheroid aggregate while EPC cells formed the outer domain (Fig. 2a, Supplementary Movie 3). Next, we tested A431 human epithelial carcinoma cells

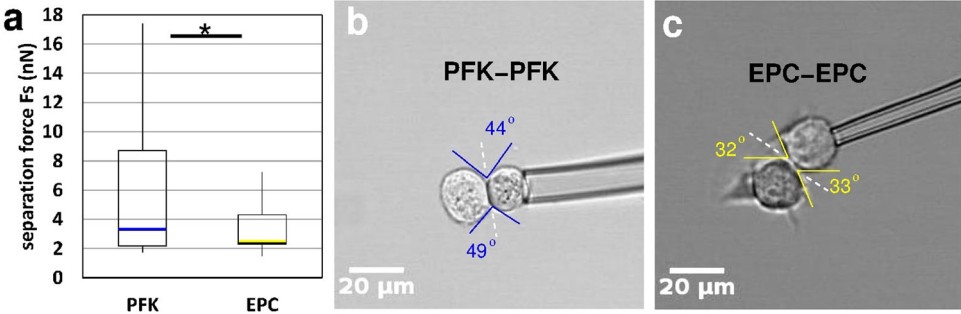

**Fig. 1 Characterization of homotypic contact strength by dual pipette aspiration assay. a** Separation forces $F_s$ of primary goldfish keratinocyte (PFK) doublets ($n = 13$) and EPC keratinocyte doublets ($n = 17$) measured after 1 min contact formation, median values are highlighted by blue and yellow lines, asterisk (*) indicates statistically significant difference with Student's $t$-test, $p < 0.05$. **b, c** Representative images of PFK doublets (**b**) or EPC doublets (**c**) held with a pipette during contact formation. Note the difference in contact angles highlighted by blue or yellow lines as a guide to the eye. Scale bar: 20 μm.

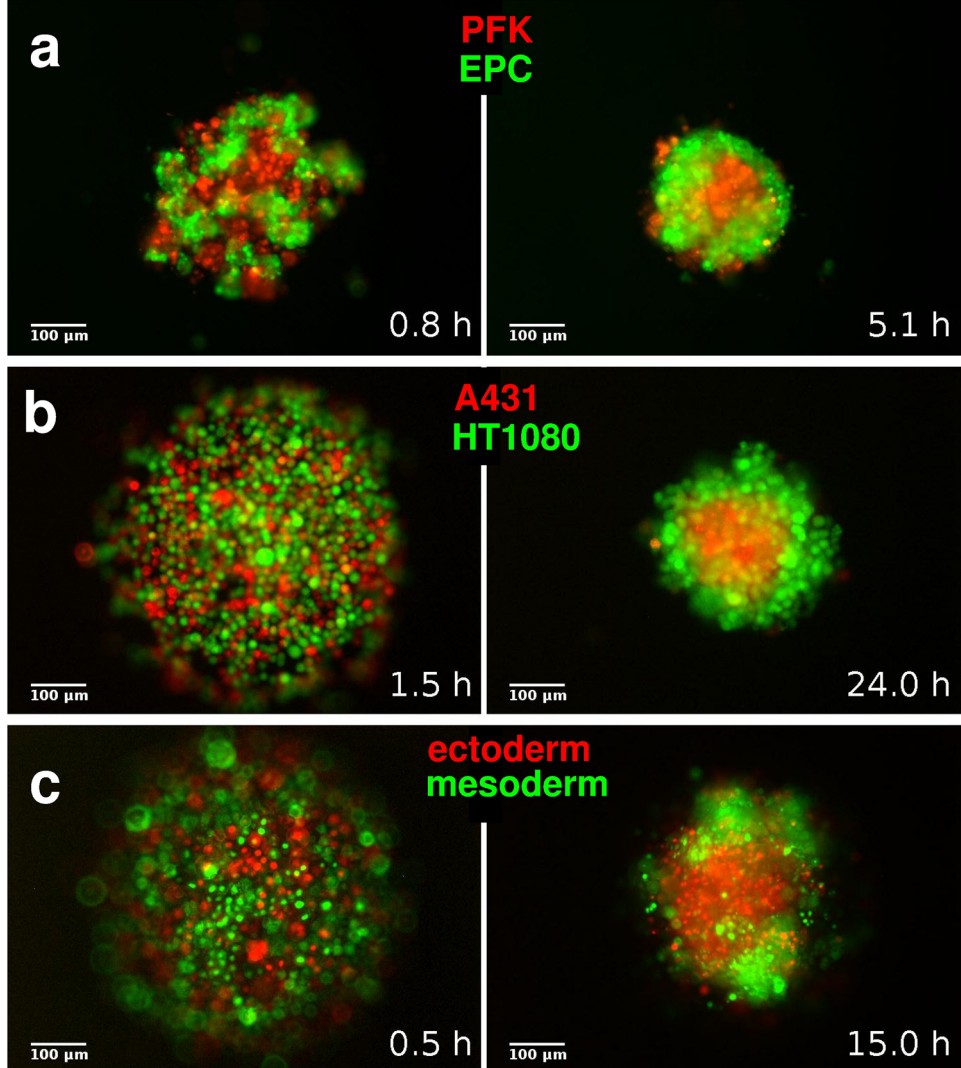

**Fig. 2 3D segregation of model cell types.** Representative epifluorescent images from image series of segregation experiments. **a** Segregation of primary goldfish keratinocytes (PFK, red) and EPC fish keratinocytes (EPC, green). **b** 3D segregation of human A431 epithelial carcinoma (A431, red) and human HT1080 fibrosarcoma (HT1080, green) cells. **c** Segregation of zebrafish ectoderm (red) and mesoderm (green) cells. Left panels in (**a–c**) show the initial phase of segregation, and the right panels show the final spatial configuration of segregated homotypic domains. Time after the initial mixing of heterotypic cell suspensions is indicated at lower right corners, scale bars: 100 μm. Also, see Supplementary Movies 3–5.

and HT1080 human fibrosarcoma cells in mixtures and observed their segregation in aggregates ($n > 20$) until reaching the final spatial configurations of homotypic domains. Segregation dynamics was slower than observed for PFK + EPC fish keratinocytes, nevertheless, A431 epithelial cells were always segregated inside and HT1080 cells formed the enveloping outer layer of spheroids (Fig. 2b, Supplementary Movie 4). Finally, we used zebrafish (*Danio rerio*) progenitor cells, dissociated from induced ectoderm or mesoderm, for basic segregation experiments. Aggregates ($n > 30$) of mixed ectoderm and mesoderm cells were imaged for several hours and eventually ectoderm cells segregated to the inside while mesoderm cells to the outside (Fig. 2c, Supplementary Movie 5), in line with expectations based on earlier data showing higher homotypic contact strengths for ectoderm cells compared to mesoderm cells, measured after several minutes of contact time[23,33].

**Dynamics of segregation depend on actomyosin contractility**. Cell sorting is analogous to phase ordering (unmixing) in viscous fluids[20], where the process is driven by differences in surface tensions of non-mixing fluid components. Multicellular aggregation and segregation are driven by tissue surface tension, influenced by cells' cortical tension resulting from cytoskeletal activity, mainly actomyosin contractility.

First, to provide a quantitative analysis of segregation dynamics we developed analysis tools to measure the size of forming homotypic cell clusters during 3D segregation of the model cell types. Using structured illumination microscopy we made z-series of fluorescent optical sections across multicellular spheroid aggregates where the cell types were selectively visualized and identified by their respective fluorescent color. We used these 2D optical sections taken with z-axis resolution of 10 microns, roughly corresponding to cell size, to reconstruct 3D structures of the segregating cell clusters and measure the size of emerging 3D clusters of the identified cell types. Cluster sizes were calculated on the basis of two-point correlation method applied within and between the 2D sections and yielding characteristic length data (see details in Methods). Optical sectioning was performed consecutively, yielding time series of cluster size data and time-lapse videos of the reconstructed 3D structures. After initial mixing, as homotypic cell domains continuously coalesce and fuse we see a characteristic increase in average cluster sizes until the final spatial configuration is reached within different durations specific for each cell type pair. Subsequently, cluster sizes tend to reach a plateau (Fig. 3a, c, e: left panels).

Next, we interfered with actomyosin contractility by treating the segregating cell mixtures with Y27632, a selective pharmacological inhibitor of Rho kinase (ROCK), the main activator of the contractile function of NM2. For the different cell type pairs we used different inhibitor concentrations ranging from 50 to 100 μM, based on earlier data[34]. As expected, general ROCK inhibition leads to delayed segregation, characterized by lower rates of cluster size growth for all three model cell type pairs, showing different sensitivities for inhibition (Fig. 3a, c, e: right panels). Nevertheless, the final spatial configuration of emerging homotypic domains was similar to those observed for untreated segregation experiments. The cell types normally segregating inside still tended to take the inner positions under ROCK inhibition while the fusion of emerging inner clusters into a single central cluster generally did not occur within the normal timeframe (Fig. 3b, d, f, Supplementary Fig. 1, Supplementary Movies 6–8).

**Inhibition of non-muscle myosin 2 assembly drives defective aggregation**. Cell aggregation in suspension and formation of spheroid aggregates depend on cells' cortical tension generated by actomyosin contractility as actin filaments are stretched and pulled by bipolar minifilaments of non-muscle myosin 2. This requires, among others, the self-assembly of NM2 into minifilaments as NM2 alone or even in dimeric state cannot move actin filaments. Multimerization of NM2A, an isoform of NM2, can be regulated by binding of S100A4 protein[35], preventing NM2 assembly into functional multimers or facilitating the disassembly of existing filaments, thereby controlling cell contractility.

To inhibit actomyosin contractile function in a cell type specific manner we used A431 cells known to express the NM2A isoform[36]. We overexpressed wild type S100A4 in a subclone of A431 cells (termed A431-S100A4). For comparison, we overexpressed an inactive truncated mutant isoform of S100A4 (MutS100A4), eliciting no effect on NM2 assembly, in another A431 subclone (termed A431-ctrl) used as negative control[34]. Western blots indicate stable expression of the wild type or mutant S100A4 variants in the A431 subclones (Supplementary Fig. 2).

We used these A431 subclones to study the effect of decreased actomyosin contractility on cell aggregation. We placed suspensions of an equal number of cells from each A431 subclone separately in non-adherent aggregation chamber wells and observed the formation of spheroid aggregates by time-lapse videomicroscopy (Supplementary Movie 9, Supplementary Fig. 3). For quantitative analysis of aggregation, we measured the perimeter of spheroids as the perimeter of their 2D projection in the image series (Fig. 4). In the case of A431-ctrl (negative control) cells, aggregation proceeded normally, characterized by decreasing perimeters, yielding round spheres with smooth surface just like normal A431 cell spheroids. As a contrast, A431-S100A4 cells with decreased contractility were lagging behind in compacting throughout the aggregation process yielding less compact and larger spheroids, characterized by higher average perimeters and berry-like rough surface, a known phenotype indicating decreased tissue surface tension[37,38].

**Selective inhibition of non-muscle myosin 2 assembly leads to inverted segregation**. During the 3D segregation of cells in heterotypic cell mixtures, homotypic domains form and grow by fusion, and the dynamics are influenced by tissue surface tension (TST) characteristic of the individual cell types, which depends on cells' cortical tension generated by actomyosin contractility. The emerging spatial positioning or geometry of the homotypic domains also depends on TST as the cell type characterized by higher TST gradually takes inner positions while the other one characterized by lower TST tends to remain in outer positions. Thus spatial positioning is indicative of TST of the segregating cell types relative to each other. To study how actomyosin contractility influences tissue surface tension and eventually spatial positioning during segregation we compared the segregation process in two different cell type pair mixtures in non-adherent aggregation wells. Red fluorescent-labeled HT1080 fibrosarcoma cells were mixed with either A431-ctrl cells overexpressing inactive mutant S100A4 or with A431-S100A4 cells overexpressing S100A4, which inhibits cell contractility by preventing NM2 assembly. Both A431 subclones express GFP for green fluorescent visualization. We observed 3D segregation within the emerging spheroid aggregates by time-lapse structured illumination microscopy and collected z-series of optical sections of the forming homotypic cell clusters, specifically identified by their fluorescent color. We used these optical sections to measure average cluster size and reconstruct the 3D structure to follow up the spatial positioning of the emerging domains. Along with optical sectioning, we also took conventional epifluorescent image series used for comparable visualization (Supplementary

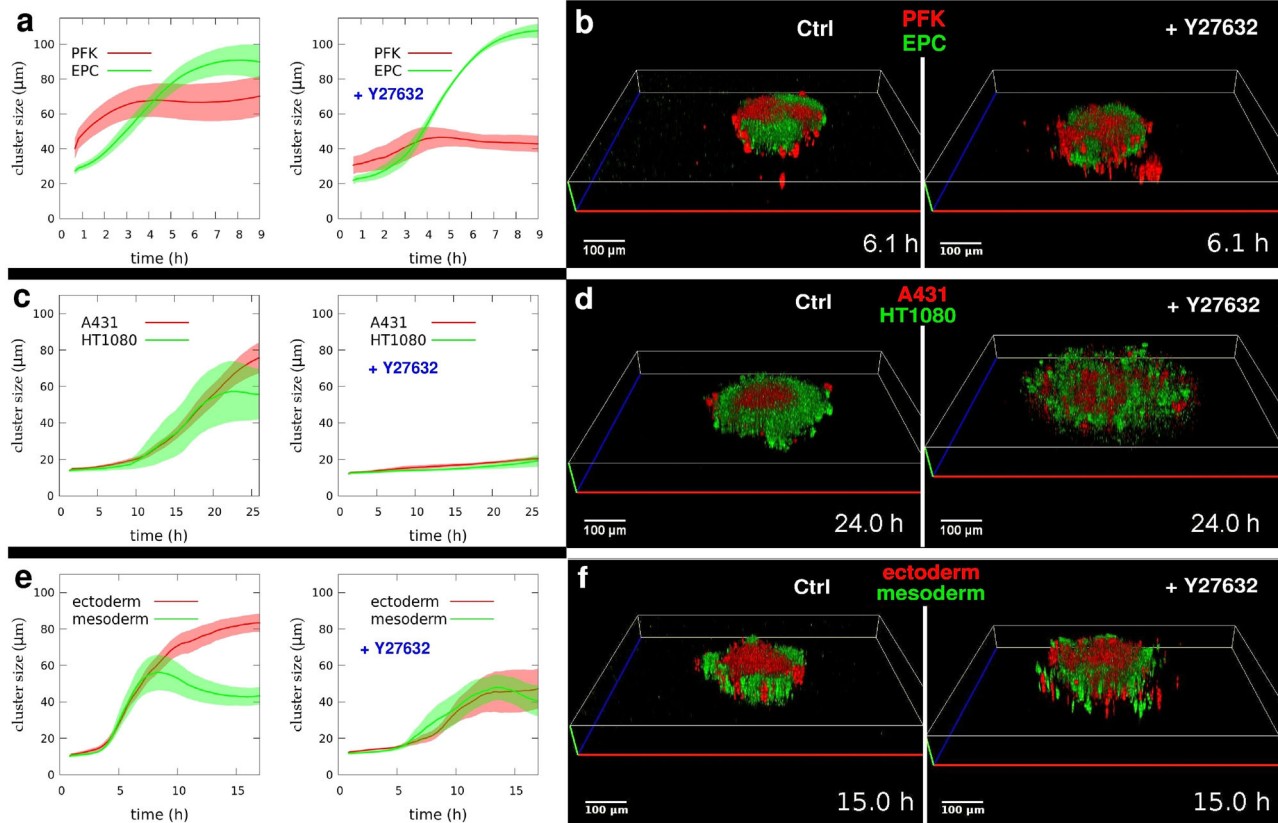

**Fig. 3 3D segregation dynamics depends on actomyosin contractility.** Quantitative analysis of segregating homotypic cell domain sizes. **a** Time-dependent growth of segregated cell domains in mixed suspensions of PKF and EPC keratinocytes untreated (left panel, $n = 7$) or treated with 100 µM Y27632 ROCK inhibitor (right panel, $n = 4$). **b** Representative 3D reconstruction images from time-lapse videos of segregating clusters of PFK (red) and EPC (green) keratinocytes after 6 h segregation in the absence (left panel) or presence (right panel) of ROCK inhibitor, see Supplementary Movie 6. **c** Analysis of segregated cell cluster sizes in A431 keratinocyte and HT1080 fibrosarcoma mixtures without (left, $n = 6$) or with 50 µM Y27632 inhibitor (right, $n = 6$). **d** Representative images of A431 (red) and HT1080 (green) segregation in the absence (left) or presence (right) of inhibitor after 24 h segregation, see Supplementary Movie 7. **e** Analysis of segregated clusters in zebrafish ectoderm and mesoderm mixtures without (left, $n = 8$) or with 100 µM Y27632 (right, $n = 6$). **f** Representative images of ectoderm (red) and mesoderm (green) segregation in the absence (left) or presence (right) of inhibitor after 15 h segregation, see Supplementary Movie 8. Error stripes represent SEM in (**a**, **c**, **e**). Time after the initial mixing of heterotypic cell suspensions is indicated at lower right corners in (**b**, **d**, **f**), scale bars: 100 µm. Also see Supplementary Fig. 1.

Movies 10 and 11). As expected from differences in aggregation dynamics of A431 epithelial carcinoma subclones with different cell contractility characteristics (Fig. 4), their segregation from HT1080 fibrosarcoma cells was also markedly different. A431 cells expressing inactive mutant S100A4 (A431-ctrl) segregated to the inside of spheroids while HT1080 cells tended to remain outside (Fig. 5a, b), with similar cluster growth dynamics and configuration of homotypic domains as seen for the segregation of normal A431 and HT1080 cells (Fig. 3d, Fig. 2b). As a marked contrast, the spatial configuration of segregated domains was inverted in the case of A431 cells expressing wild type S100A4 (A431-S100A4) as these cells showed slower cluster growth dynamics and eventually segregated to the outside of spheroids while HT1080 cells this time tended to take the inner positions (Fig. 5c, d). This inverted spatial positioning of homotypic domains indicates that stable inhibition of actomyosin contractility in A431-S100A4 cells eventually reduced their specific tissue surface tension below that of HT1080 cells (Supplementary Fig. 4).

**Constitutively active ROCK causes defective contact formation and aggregation.** Contact strength in cell coupling can be characterized by the experimentally measurable separation force ($F_s$) of cell doublets and their contact angle (Fig. 1).

Cell cortex tension, generated by cortical actomyosin contractility, plays a major role in contact formation and expansion and it was shown earlier that reduced cortical tension is required at cell-cell interface compared to cell-medium interface for effective contact expansion[23]. To study the role of this localized tuning of cortical tension in contact formation we interfered with cells' actomyosin contractility regulation. For making it easier to interpret our experiments we compiled a schematic figure summarizing: (i) current knowledge about cytoskeleton regulation by cell adhesion molecules, (ii) the major physical components generating tissue surface tension as it is currently known[2,3,23], and (iii) the expected consequences of our experimental interventions into cell contractility regulation (Fig. 6).

For a model cell type we chose zebrafish ectoderm cells characterized by high separation force ($F_s$) and high contact angle in cell doublets[23] as well as segregation positioning inside when mixed with mesoderm cells, indicating higher tissue surface tension for ectoderm (Fig. 2c, Fig. 3f). Using a *caROCK* mRNA microinjected in zebrafish embryos induced to yield ectoderm cells (see Methods for details) we overexpressed a constitutively active ROCK isoform, which constantly activates non-muscle myosin 2 (ectoderm-caROCK). For comparable negative control, we used normal ectoderm cells dissociated from ectoderm-induced embryos (ectoderm-ctrl).

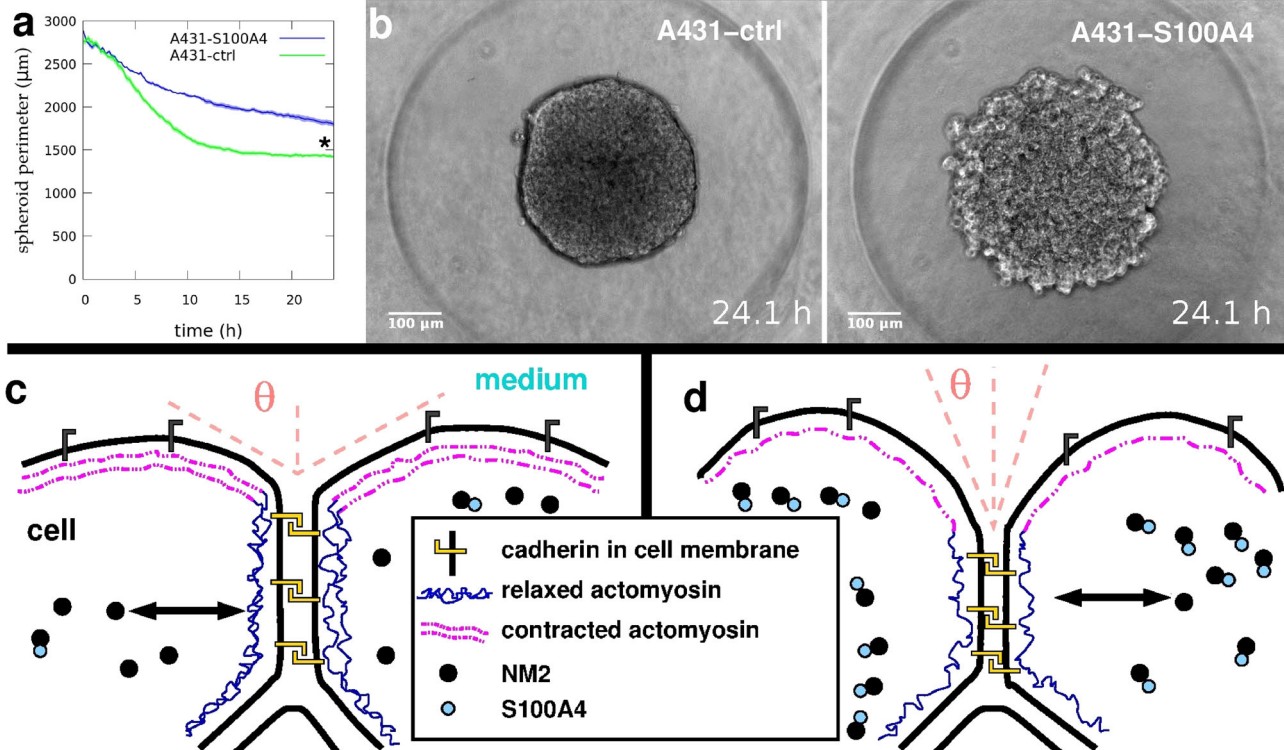

**Fig. 4 Aggregation dynamics depend on non-muscle myosin 2 assembly.** Quantitative analysis of aggregation of A431 cells. **a** Time-dependent decrease in the mean perimeter of spheroids aggregating from homotypic cell suspension of A431 cells overexpressing either NM2 assembly inhibitor S100A4 (A431-S100A4, $n = 10$) or its inactive mutant isoform (A431-ctrl, $n = 10$). Error stripes represent SEM, asterisk (*) at $t = 24$ h indicates a statistically significant difference with Student's $t$-test, $p < 0.01$. **b** Representative phase-contrast images from time-lapse videos of spheroids of A431-ctrl cells (left panel) or A431-S100A4 cells (right panel) after 24 h aggregation from suspension. Note the difference in spheroid surface roughness. Scale bar: 100 μm. Also see Supplementary Movie 9 and Supplementary Fig. 3. **c**, **d** Schematic representations of surface cells highlighting the cytoskeletal components involved in cortical tension generation. **c** Normal cells with effective multicellular compaction characterized by large contact angle Θ due to the contracted actomyosin network at the cell-medium surface and relaxed actomyosin at the cell-cell interface coupled by cadherins. Assembly of NM2 monomers into filaments is controlled by normal levels of S100A4, assembly, and disassembly processes are symbolized by the double-headed arrow. **d** Experimentally increased levels of S100A4 lead to the sequestration of NM2 monomers and shifting towards the disassembly of filaments, assumed to result in a shift towards reduced cortical actomyosin tension and reduced multicellular compaction. Definitions of symbols are shown in the central text box.

First, we measured the contact angle Θ of homotypic cell doublets formed randomly in suspensions of ectoderm-caROCK cells on non-adherent substrate in 30 min and compared them to ectoderm-ctrl cell doublets formed under similar conditions. Overexpression of caROCK significantly decreased contact angle as supported by Student's $t$-test ($p < 0.001$) performed on cell doublet data (Fig. 7).

Next, we studied the aggregation dynamics of caROCK-expressing ectoderm cells comparing them to ectoderm-ctrl cells. We placed separate homotypic cell suspensions of an approximately equal number of cells from these two cell types into non-adherent aggregation chamber wells and imaged their aggregation into spheroids by time-lapse videomicroscopy (Supplementary Movie 12). For quantitative analysis, throughout the aggregation process, we measured the perimeter of the 2D projection of spheroids in the image series and normalized time-dependent perimeter data with the initial perimeter of each spheroid. For normal ectoderm-ctrl cells, average spheroid perimeters decreased resulting in compact spheroids with smooth surfaces. Conversely, ectoderm-caROCK aggregation was slower resulting in less compact spheroids characterized by higher perimeters and rough, berry-like surfaces, an indication of decreased tissue surface tension characterizing these cells (Fig. 8a, b).

In order to verify the effect of caROCK overexpression on myosin activity we used these spheroids to immunodetect the phosphorylated form of the myosin regulatory light chain (MLC) protein. MLC is primarily phosphorylated by ROCK whereby NM2

is turned into its active state enabling it to assemble into filaments and eventually move actin filaments according to the Lymn-Taylor functional cycle of the actomyosin complex. Therefore the level and subcellular localization of phospho-MLC is indicative of actomyosin contractility. Compared to ectoderm-ctrl cells, the ectoderm-caROCK cells showed considerably increased p-MLC immunodetection levels throughout the aggregates including the bulk region, indicating increased actomyosin contractility there (Fig. 8c–f).

**Cell type specific constitutively active ROCK causes inverted segregation.** The spatial configuration of homotypic cell domains formed in the course of 3D segregation in heterotypic cell mixtures is indicative of tissue surface tensions characterizing the specific cell types. Cell-cell contact formation requires localized reduction of cortical tension at cell-cell interface, which can be achieved by localized inhibition of actomyosin contractility, a hypothesis that we intended to test.

To study the role of localized inhibition of actomyosin contractility at cell-cell interfaces we used zebrafish ectoderm cells overexpressing a constitutively active ROCK isoform (ectoderm-caROCK), overriding such localized inhibition. Ectoderm-caROCK cells are characterized by decreased aggregation properties and reduced tissue surface tension, as compared to normal ectoderm-ctrl cells (Fig. 8).

We mixed zebrafish mesoderm cells in equal amounts with either ectoderm-ctrl cells or ectoderm- caROCK cells in non-

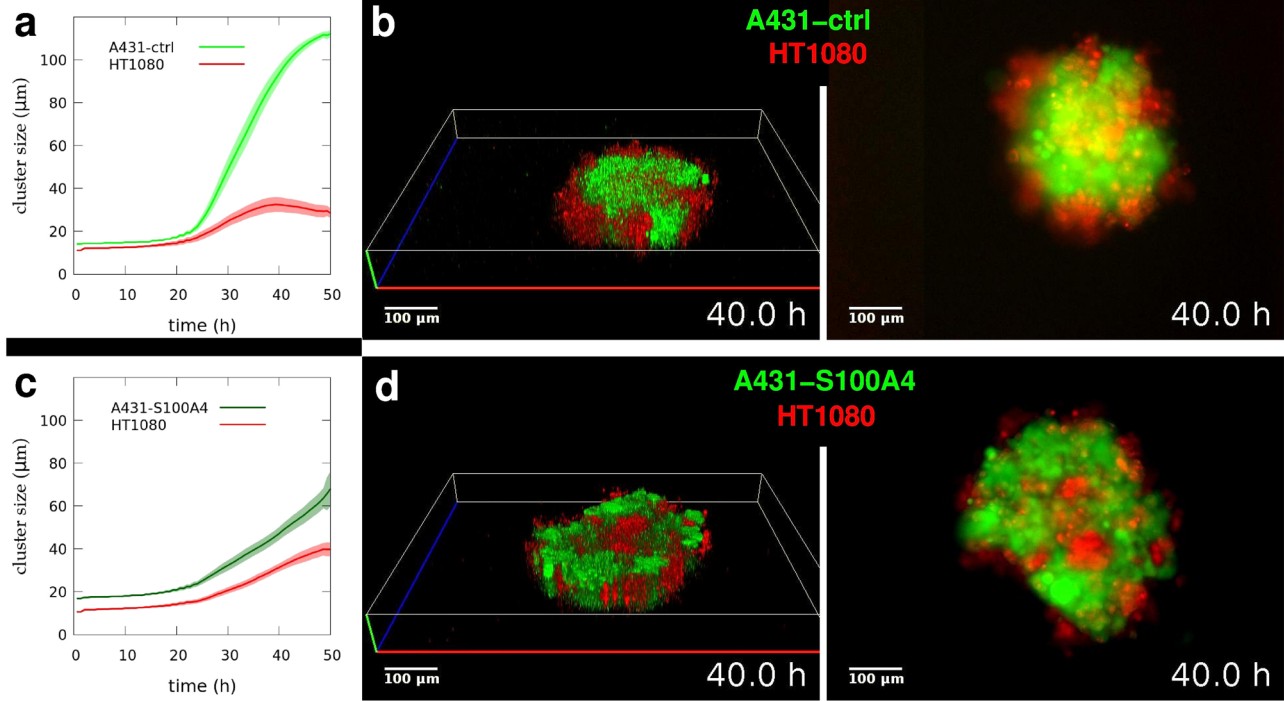

**Fig. 5 Spatial positioning during segregation depends on non-muscle myosin 2 assembly and function. a** Time-dependent growth of segregated cell domains in mixed suspensions of A431 keratinocytes overexpressing inactive mutant S100A4 (A431-ctrl) and HT1080 fibrosarcoma cells ($n = 6$).
**b** Representative 3D reconstruction (left panel) and simultaneous epifluorescent (right panel, bottom view) images from time-lapse videos of segregating clusters of A431-ctrl (green) and HT1080 (red) cells after 40 h of segregation, see Supplementary Movie 10. **c** Time-dependent growth of segregated domains in mixtures of A431 keratinocytes overexpressing NM2 assembly inhibitor S100A4 (A431-S100A4) and HT1080 fibrosarcoma cells ($n = 6$).
**d** Representative 3D reconstruction (left panel) and epifluorescent (right panel, bottom view) images of segregating clusters of A431-S100A4 (green) and HT1080 (red) cells after 40 h of segregation, see Supplementary Movie 11. Note the inverted configuration of segregated domains here, compared to (**b**). Scale bar: 100 μm in (**b**, **d**). Error stripes represent SEM in (**a**, **c**). See also Supplementary Fig. 4.

adherent aggregation wells and observed their segregation by time-lapse structured illumination microscopy combined with conventional epifluorescence microscopy (Supplementary Movies 13 and 14). Throughout the segregation process, we collected z-series of optical sections of the specifically identified homotypic cell clusters to measure cluster sizes as well as to reconstruct 3D structure revealing the spatial configuration of the emerging homotypic domains.

Ectoderm-ctrl cells segregated to the inside of spheroids leaving mesoderm cell clusters in outer positions (Fig. 9b), while cluster growth dynamics and spatial configuration were similar to other tests with these cell types (Fig. 2c, Fig. 3e). As a marked contrast, segregation configuration of ectoderm-caROCK cells with mesoderm cells was inverted (Fig. 9d). Mesoderm cells now tended to segregate to the inside of spheroids with normal cell cluster growth dynamics, while ectoderm-caROCK cluster growth rate was steadily reduced (Fig. 9c), and ectoderm-caROCK clusters segregated to the outside as an indication of decreased tissue surface tension (Supplementary Fig. 5). This is consistent with decreased tissue surface tension observed for ectoderm-caROCK cells in aggregation (Fig. 8) and supports the claim that constitutively active ROCK prevents local inhibition of actomyosin contractility at the cell-cell interface that is normally required for effective contact formation and expansion and eventually generation of high tissue surface tension.

## Discussion
The main mechanisms acting while a fertilized egg develops into a fully developed embryo are still not completely understood, but some of the ingredients of the process can be identified as being

dominated by diffusion, spontaneous collective migration, and migration as a result of some guiding signals (e.g. gradients of chemokines and/or growth factors)[18]. In light of the above considerations we were motivated to investigate two aspects of the emergence of morphology during 3D cell segregation: (i) we tested whether the conclusions of prior studies about the pattern formation of segregating zebrafish progenitor cells apply to a wider range of vertebrate cells and (ii) we applied a computational analysis to the time lapse 3D recordings, which opens the way for analyzing the relative roles of the diffusion and group motion mechanisms during segregation.

The two modalities – diffusion and collective motion – result in distinct characteristic time dependencies. During the segregation of two types of cells of about the same number, theory predicts that the characteristic linear size of homotypic cell clusters grows with time as power law according to $t^z$ with $z = 1/3$, while the coherent motion of cells of the same type results in a much faster pace of segregation characterized by a linear growth of the cluster diameters with $z = 1$, accordingly[39]. Since one of the focuses of our study is the time dependence of the level of segregation, it was essential that we analyzed our measured cell configurations by a well-established method borrowed from statistical physics and relying on determining the so-called density correlation function. Our quantitative data on the dynamics of various in vitro models of 3D cell segregation can be used by future studies on computational modeling of 3D segregation.

Tissue surface tension (TST) is an important biomechanical property characterizing multicellular organization and is required for tissue self-organization. Previous studies have implicated that interface-specific regulation of cortical tension is required for cell-

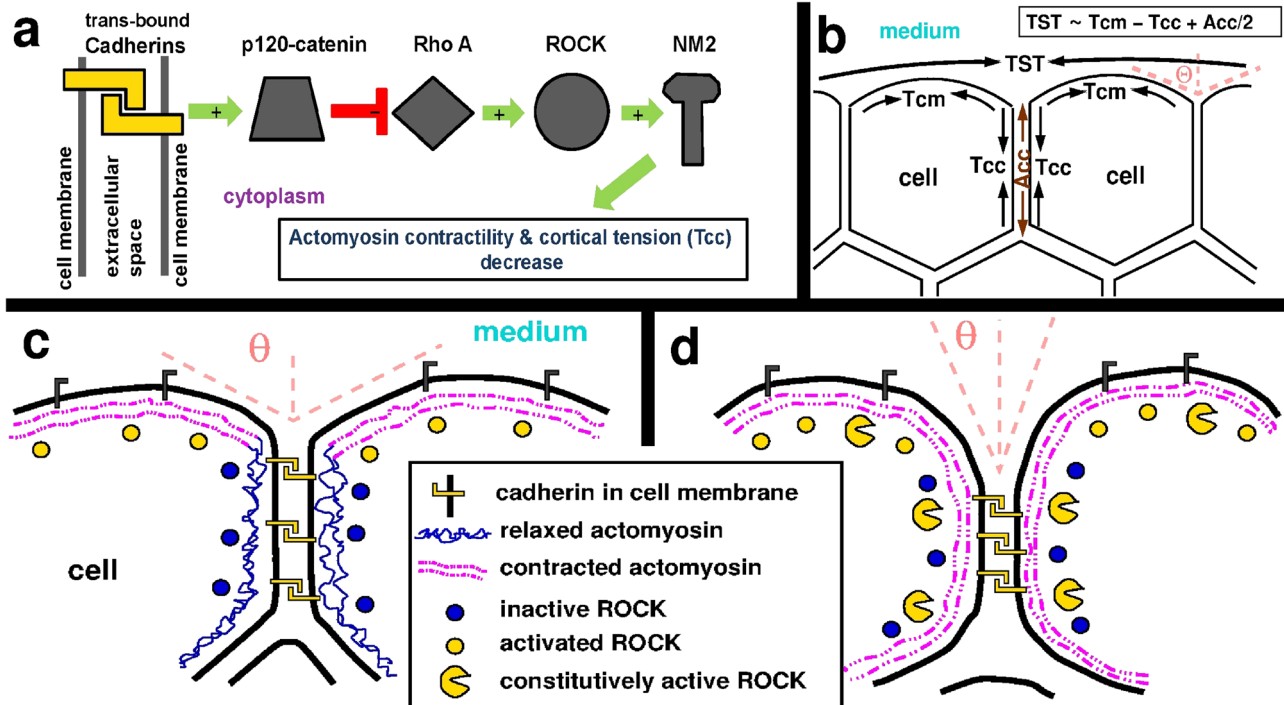

**Fig. 6 Schematic summary of experimentally studied aspects of tissue surface tension regulation. a** Signaling from cell adhesion molecules to the cytoskeleton. Trans-binding of cell surface cadherins of contacting cells results in decreased actomyosin contractility and cortical tension at the cell-cell interface (Tcc). Green arrows indicate activation, and the red symbol represents inhibition. A detailed description of this signaling cascade and references are included in the Discussion. **b** Summary of tissue surface tension (TST) generation by cells in an aggregate, shown as a usual schematic representation. Cell cortical tension at the cell-medium interface (Tcm) and adhesion tension (Acc) contribute positively to TST whereas cortical tension at the cell-cell interface (Tcc) has a negative contribution. Tensions are represented by arrows. A contact angle Θ is indicated by dotted lines as a guide to the eye. **c**, **d** Schematic images of cell aggregates with key components of cortical tension regulation highlighted by symbols to explain the experimental interventions. **c** Normal cells at the surface of the aggregate showing effective multicellular compaction characterized by large contact angle Θ due to a relaxed actomyosin network at the cell-cell interface and low cortical tension as a result of signaling from trans-bound cadherins. **d** Genetically manipulated cells expressing the constitutively active ROCK isoform, which is expected to constantly activate actomyosin contractility at the cell-cell interface regardless of inactive endogenous ROCK here. The proposed impact is that caROCK maintains higher cortical tension at the cell-cell interface, leading to reduced TST and less effective compaction characterized by a smaller contact angle. Definitions of symbols are shown in the central text box.

cell contact formation and it involves active reduction of cortex tension at cell-cell interfaces. During cell contact formation and aggregation, the localization of non-muscle myosin 2 (NM2) changes at interfaces: it accumulates at cell-medium interfaces, whereas it is depleted at cell-cell interfaces, implying that interface-specific cortical tension regulation involves interface-specific recruitment of NM2[21,23]. The contact strength of two contacting cells can be characterized by the separation force, which is experimentally measurable ex vivo by the dual pipette aspiration (DPA) assay. Besides the separation force, the contact angle characterizing the contacting cells is also indicative of the contact strength as higher contact angles coincide with higher separation forces. Contact strength is dependent on cell surface adhesion molecules, contributing positively, and also limited by their anchorage to the cytoskeleton, while a negative contribution comes from the tension of the cell cortex at the site of contact. Cortical tension at the contact is actively reduced during contact formation of germ layer progenitor cells, running along with a reduction of NM2 at the contact area[23].

Studies in epithelial cultures have revealed regulation mechanisms through which cadherin transbinding at cell-cell interface initiates signaling towards the cytoskeleton[26,40]. The implied mechanism involves p120-catenin recruitment to cadherin complexes[28], where p120-catenin inhibits RhoA activity locally[27] through the recruitment of p190RhoGAP[30]. Inhibition of RhoA leads to further downstream inactivation of ROCK and

eventually downregulation of NM2 activity. Alpha-catenin contributes to the binding and functions of p120-catenin[41] and the importance of this interaction is indicated by studies showing that loss of alpha-catenin in embryonic cell aggregates results in reduced tissue surface tension[42]. The signaling cascade starting from trans-bound cadherins eventually downregulates NM2-based actomyosin contractility, enabling local control of cortical tension (Fig. 6). NM2 can also act as a mechanosensor since mechanical tension stabilizes its association with actin[25,43] and thus recruitment of NM2 to areas of high cortical tension can provide additional means of site-specific regulation of the cortical cytoskeleton.

In this study, we have applied several ways to perturb actomyosin contractility and analyzed the 3D segregation of several model cell type pairs as a functional read-out to study the effect of such perturbations. We show that general inhibition of ROCK by the cell permeable inhibitor Y27632 decreases TST and as a result delays and slows down segregation, however the final spatial configuration of segregated domains is little affected (Fig. 3). Cell types characterized by high TST and segregating inside under normal conditions continued to do so under general ROCK inhibition, indicating that inhibition of NM2 activity in both segregating cell types does not considerably change the difference between their respective TST. This may be due to the fact that in the lack of considerable NM2-based cortical tensions, the unperturbed differences in adhesion tension can now dominate

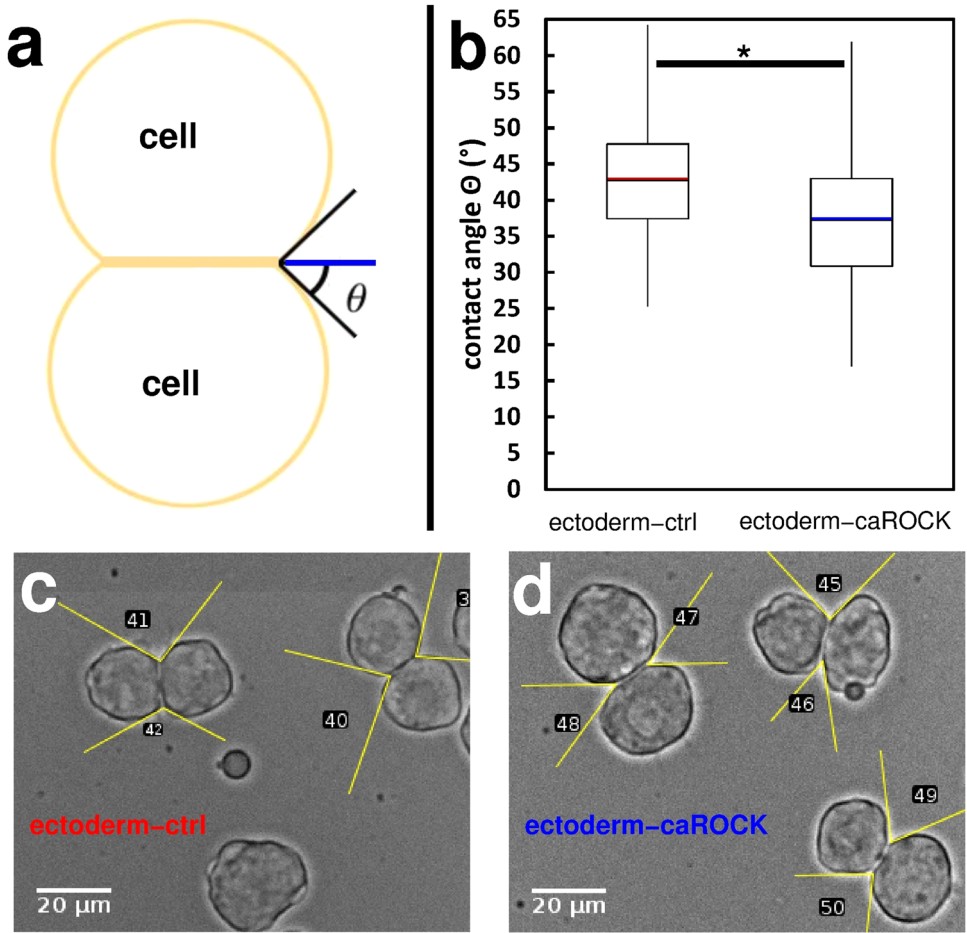

**Fig. 7 Contact formation is influenced by the regulation of actomyosin contractility. a** Schematic image of a homotypic cell doublet after contact formation. Contact angle Θ is indicated by lines as a guide to the eye. **b** Quantitative analysis of contact angles of homotypic doublets of untreated ectoderm cells (ectoderm-ctrl, $n = 170$) or ectoderm cells overexpressing constitutively active ROCK (ectoderm-caROCK, $n = 210$). Contact angles were measured after 30 min of contact formation. Median values are highlighted by red and blue lines, asterisk (*) indicates a statistically significant difference with Student's t-test, $p < 0.001$. **c, d** Representative phase-contrast images of freely adhering homotypic cell doublets in ectoderm-ctrl (**c**) or ectoderm-caROCK (**d**) suspensions after 30 min. Note the difference between (**c**) and (**d**) in contact angles highlighted by yellow lines as a guide to the eye. Numbers are contact angle IDs for analysis. Scale bar: 20 µm.

TST generation and drive slower segregation. An alternative interpretation is that the applied concentrations of the ROCK inhibitor, although falling in the high range for the studied cell types[34], can allow cell type-specifically different residual NM2 activities accounting for different cortical tensions resulting in the unchanged segregation configurations.

Quantitative analysis of the dynamics of segregation can reveal here the sensitivity of the dynamics to general inhibition of actomyosin contractility and this can also reveal the different contributions of cortical tensions and adhesion tension to TST generation. The rate of cluster size growth in keratinocyte segregation is little affected under actomyosin inhibition (Fig. 3a), indicating the higher contribution of adhesion to TST generation, as compared to the segregation of A431 and HT1080 cells where inhibition of actomyosin contractility profoundly slows down segregation (Fig. 3c), indicating the higher contribution of cortical tensions to TST. Further tests dissecting the contributions of actomyosin contractility and cadherin- based adhesion tension to TST[31] would be required to more specifically interpret this phenomenon, which clearly falls outside the scope of this paper.

We demonstrate that cell type-specific inhibition of actomyosin contractility through the inhibition of NM2 assembly into functional filaments by S100A4 reduces TST characterizing that cell type. This reduction is manifested in segregation configurations generated by the two human tissue cell types where formerly inside-positioned epithelial cells are now segregated outside owing to their shortage in functionally assembled NM2 filaments (Fig. 5). Inhibition of NM2 assembly throughout the cell also results in reduced compaction of homotypic cell aggregates and lack of stretched cells at the surface of the forming spheroids (Fig. 4) indicating that generation of cortical tension at cell-medium interface, which is the main drive of the aggregation process, clearly depends on functional actomyosin structure. Note here that S100A4 inhibits only the NM2A isoform[35], and the different NM2 isoforms may have distinct roles in the regulation of cortical tension, however, this does not contradict our conclusion that TST can be reduced by inhibiting NM2 assembly. A similar reduction in TST was demonstrated earlier[21] upon ubiquitous inactivation of NM2 by the expression of a dominant negative ROCK isoform in zebrafish ectoderm cells, suggesting a universal mechanism in cell types of various origins.

Finally and importantly, we show that if the interface-specific regulation of cortical tension is overridden by expressing a constitutively active ROCK mutant in zebrafish ectoderm cells, a cell type normally characterized by high TST and segregating inside from mesoderm cells, these ectoderm cells now segregate outside,

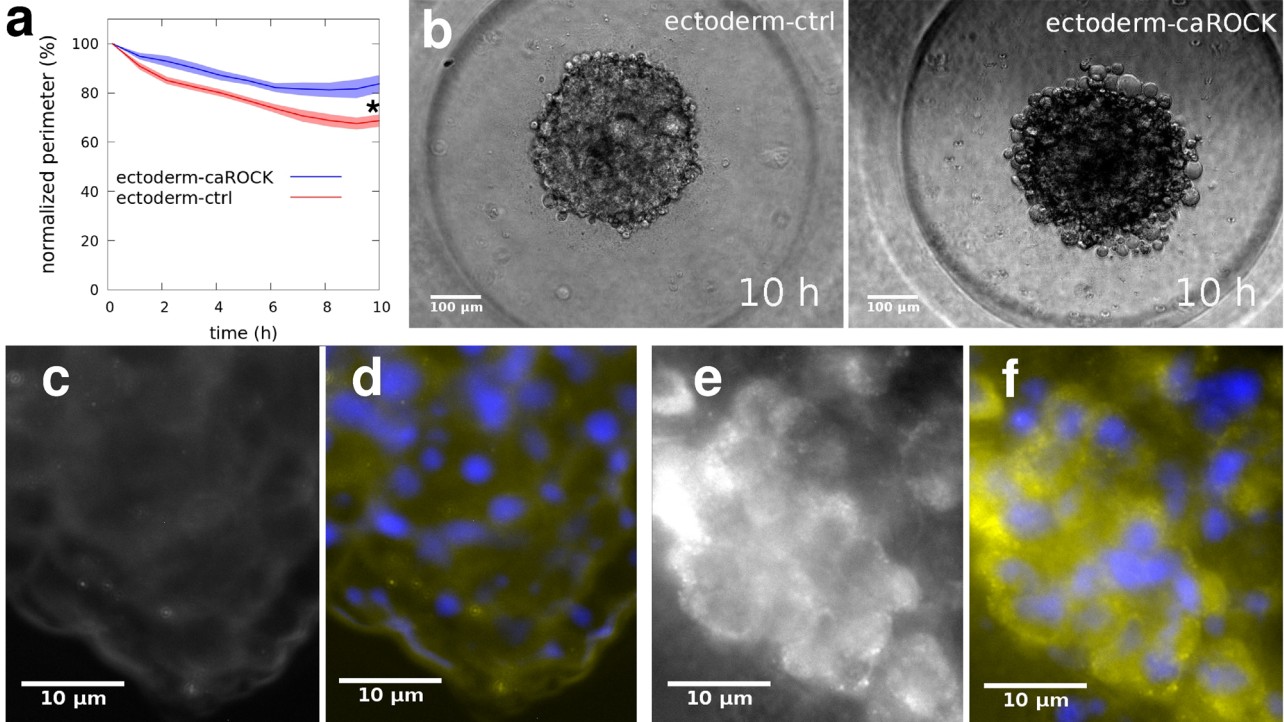

**Fig. 8 Aggregation dynamics is influenced by regulation of actomyosin contractility.** Quantitative analysis of aggregation of zebrafish ectoderm cells. **a** Time-dependent decrease in the mean perimeter of spheroids aggregating from homotypic cell suspensions of either untreated ectoderm cells (ectoderm-ctr, $n = 20$) or ectoderm cells overexpressing constitutively active ROCK (ectoderm-caROCK, $n = 20$). Spheroid perimeters are normalized with and plotted as a percentage of initial perimeters, error stripes represent SEM, asterisk (*) at $t = 10$ h indicates a statistically significant difference with Student's $t$-test, $p < 0.01$. **b** Representative phase-contrast images from time-lapse videos of spheroids of ectoderm-ctrl cells (left panel) or ectoderm-caROCK cells (right panel) after 10 h of aggregation. Note the difference in spheroid surface roughness. Scale bar: 100 µm. Also, see Supplementary Movie 12. **c–f** Immunofluorescent detection of phospho-myosin light chain (p-MLC) in spheroids of ectoderm cells. **c**, **d** Ectoderm-ctrl spheroid with only p-MLC labeling (**c**) or merged image of p-MLC (yellow) and NucBlue (blue) labels. **e**, **f** A spheroid of ectoderm-caROCK cells labeled for p-MLC (**e**) or merged image of p-MLC label (yellow) and NucBlue (blue) staining. Cell nuclei are visualized by NucBlue staining. Note that the p-MLC immunofluorescence signal is hardly seen in (**c**) while it is much more pronounced after the introduction of caROCK in (**e**). Scale bar: 10 µm in **c–f**.

indicating reduced TST (Fig. 9). Slightly delayed and reduced compaction during multicellular aggregation of caROCK-expressing ectoderm cells is also characteristic along with the appearance of loosely adhering round cells at the surface of these spheroids, indicating a tendency of reduced TST, while the activity of NM2, assessed by immunofluorescence, was also elevated throughout these spheroids (Fig. 8). Importantly, when homotypic ectoderm cell doublets are formed in culture by cell-cell adhesion, we observe a tendency of reduced contact angles in caROCK-expressing cell doublets (Fig. 7), which is an indication of increased cell-cell interfacial tension here.

All these observations indicate that caROCK, which cannot be downregulated, constantly activates NM2 and thus keeps actomyosin contractility high at all sites of the cell cortex, while cortical tension should normally be reduced at the cell-cell interface due to the inactivation of endogenous ROCK by signaling from trans-bound cadherins here. Conversely, due to a lack of trans-bound cadherins at the cell-medium interface, endogenous ROCK is not inactivated there by this signaling pathway and keeps actomyosin contractility high, which can be further increased by caROCK. Because TST primarily depends on the difference of cortical tension at the cell-medium interface (Tcm) and cortical tension at the cell-cell interface (Tcc) with minor contribution of cell-cell adhesion tension (Acc) as TST ≈ Tcm-Tcc+Acc/2 in general[2,3,23,42], this difference is diminished in caROCK-expressing cells due to their decreased ability to reduce cortical tension at the cell-cell interface, while the minor contribution of cell-cell adhesion tension is not assumed to change (Fig. 6).

The above findings, when compared to earlier studies[21,23], jointly provide experimental proof that interface-specific tuning of cortical tension through the local regulation of the actomyosin contractile system by ROCK determines TST of multicellular structures, which drives their self-organization, including segregation. Cell type-specific interventions in the regulation of NM2 activity or NM2 assembly into functional filaments can be used as potential tools to modulate TST and eventually tissue self-organization in the emerging fields of tissue engineering.

## Methods

**Animals**. Wild type (AB) zebrafish (*Danio rerio*) used in this study were maintained and bred in the fish facility of ELTE Eötvös Loránd University according to standard protocols[44,45]. All protocols used in this study were approved by the Hungarian National Food Chain Safety Office (Permit Number: XIV-I-001/515-4/2012). Goldfish (*Carassius auratus*) used in this study were maintained at the Department of Biological Physics of Eötvös Loránd University in standard aquarium conditions, approved by University Institutional Animal Care and Use Committee.

**Plasmids and mRNA**. The pCS2+cyclops, pCS2+lefty1, pCS2+mCherry-H2A, pCS2+EGFP-H2B plasmids were used for producing mRNA for microinjections. Constitutively active ROCK (ca-ROCK) CDS was also cloned into pCS2+ backbone from pCAG-myc-p160Δ3 construct[46], a kind gift from Jeffery Amack (State University of New York Upstate Medical University). Plasmids were linearized with KpnI restriction enzyme and were transcribed in vitro with the mMessage mMachine SP6 kit (Ambion, AM1340) according to the manufacturer's protocol.

**Cell cultures**. Cell cultures made of primary cells or commercially available cell lines were kept in Dulbecco's Modified Eagle Medium (DMEM, Sigma D6429)

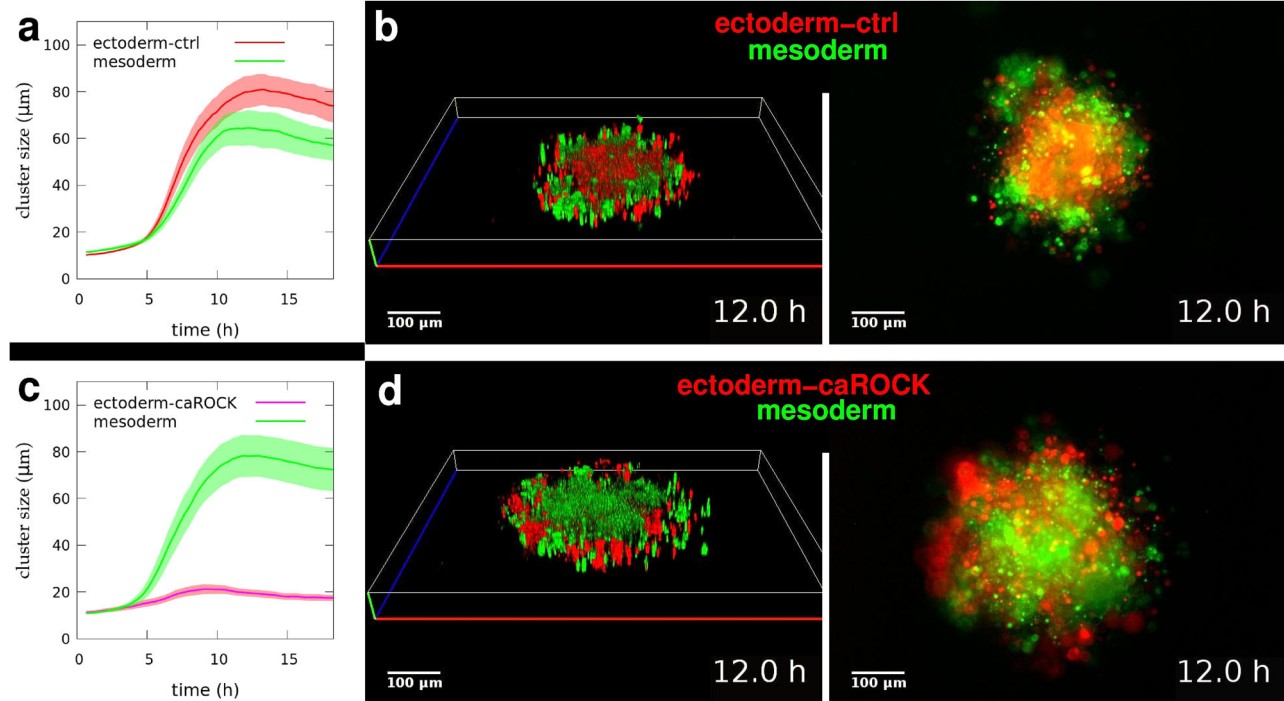

**Fig. 9 Spatial configuration of segregated domains depends on actomyosin contractility regulation. a** Time-dependent increase in mean segregated cell cluster sizes in aggregates ($n = 16$) forming in mixed suspensions of normal zebrafish ectoderm cells (ectoderm-ctrl) and mesoderm cells. **b** Representative 3D reconstruction (left panel) and simultaneous epifluorescent (right panel, bottom view) images from time-lapse videos of segregating clusters of ectoderm-ctrl (red) and mesoderm (green) cells after 12 h of segregation, see Supplementary Movie 13. **c** Time-dependent growth of segregated cell cluster sizes in aggregates ($n = 15$) of mixed ectoderm cells overexpressing constitutively active ROCK (ectoderm-caROCK) and mesoderm cells. **d** Representative 3D reconstruction (left) and epifluorescent (right, bottom view) images of segregating clusters of ectoderm-caROCK (red) and mesoderm (green) cells after 12 h of segregation, see Supplementary Movie 14. Note the inverted spatial configuration of segregated domains here, compared to panel (**b**). Scale bar: 100 µm in (**b, d**). Error stripes represent SEM in (**a, c**). Also see Supplementary Fig. 5.

containing 4 mM L-glutamine, supplemented with 10% fetal calf serum (FCS, GIBCO) and Penicillin-Streptomycin-Amphotericin B (Lonza), in a humidified incubator with 5% $CO_2$ atmosphere in tissue culture grade 6-well plates or culture dishes (Greiner) at 37 °C or 25 °C, depending on the cell type. Cell cultures established from dissociated zebrafish embryos were kept for experiments in $CO_2$-independent Medium (GIBCO) supplemented with 10% FCS (GIBCO), 4 mM L-glutamine and Penicillin-Streptomycin-Amphotericin B (Lonza) in a humidified incubator at 28 °C.

Primary fish keratinocytes were isolated from scales of goldfish (*Carassius auratus*) as described earlier[47]. Briefly, without sacrificing the goldfish, a few scales were isolated and kept in DMEM-10%FCS culture medium for one day while keratinocytes migrated out of the scales to the culture dish. EPC fish keratinocyte cell line was purchased from ATCC (CRL-2872) and kept in DMEM-10%FCS medium.

Cell suspensions were prepared from both keratinocyte cultures after brief incubation in PBS (phosphate buffered saline, 0.1 M phosphate, 0.9% NaCl, pH 7.4, GIBCO) and transferred in $CO_2$-independent Medium-10% FCS at 25 °C for experiments.

A431 human epithelial carcinoma cell line (CRL-1555) and HT1080 human fibrocarcinoma cell line (CCL-121) were obtained from ATCC and cultured in DMEM-10%FCS at 37 °C. A431 or HT1080 monolayer cultures were briefly incubated with 0.5 mg/ml trypsin and 0.2 mg/ml EDTA (Sigma) in PBS (GIBCO) then cells were rinsed and transferred in suspensions in a culture medium for experiments.

Zebrafish (*Danio rerio*) embryos were induced to consist of only one germ layer progenitor cell type by microinjecting 1-cell stage embryos with *lefty1* mRNA (100 pg) for ectoderm or *cyclops* mRNA (100 pg) for mesoderm. Fluorescent labeling was achieved by co-injecting at 1-cell stage with mRNA (200 pg) encoding either histone2A-mCherry resulting in red-labeled nuclei in ectoderm or histone2B-EGFP resulting in green nuclei in the mesoderm. To create ectodermal cells with altered Rho kinase activity, we co-injected lefty1 and EGFP-H2A mRNAs with *caROCK* mRNA (100 pg). Embryos were kept in L15 medium (Sigma, L1518) at 28.5 °C until sphere stage (4 hpf) then they were mechanically dissociated into single cell suspensions in $CO_2$-independent Medium-10%FCS at 28 °C for experiments.

**Cell labeling**. Unlabeled cells were labeled with fluorescent dyes for fluorescence microscopy imaging. Generally, all cell types or cell lines were labeled by

incubation with the dyes in monolayer culture conditions in DMEM-10% FCS culture medium. Primary goldfish keratinocytes and A431 human epithelial carcinoma cells were labeled with 10 µM CellTracker Red CMTPX (Invitrogen, C34552) for 90 min or 30 min, respectively. EPC fish keratinocytes and HT1080 human fibrosarcoma cells were labeled with 5 µM CellTracker Green CMFDA (Invitrogen, C7025) for 90 min or 30 min, respectively.

**Immunolabeling**. Zebrafish ectoderm cell aggregates were fixed in 4% paraformaldehyde in PBS for 30 min and per- meabilized in 0.1% Triton X-100 in PBS for 10 min. The non-specific binding sites were blocked by incubation in a culture medium containing 10% serum for 2 h. Primary antibody against phosphorylated (pS19) myosin light chain (rabbit, polyclonal, Anti-MYL12A phospho S19, Abcam ab2480) was applied in 1/100 dilution for 2 h at room temperature and then overnight at 4 °C. Anti-rabbit secondary antibody conjugated with AlexaFluor-555 (Southern Biotech, 4030-32) was used in 1/200 dilution for 4 h at room temperature. All incubations were followed by triple washing steps in PBS for 1 h. Finally, immunolabeled aggregates were mounted on microscopic slides (Thermo Scientific) using a mounting medium (Prolong Glass Antifade Mountant with NucBlue Stain, Invitrogen, P36981) containing NucBlue counterstain to visualize cell nuclei. Fluorescent labels were imaged using a Zeiss Axio Observer Z1 microscope with Zeiss EC Plan-Neofluar 40x/0.75 or Olympus A 100x/1.3 objectives and Zeiss AxioCam MRm CCD camera. Images were processed using NIH ImageJ software.

**Aggregation chambers**. Aggregation microwell chambers were made by casting molten liquid 2% agarose (Invitrogen) in PBS and allowing it to gelate in a PDMS negative form (3D Petri Dish, Microtissues) containing pillars that define non-adherent wells with a diameter and depth of 800 µm. After removing from the negative form, the agarose gel chambers were equilibrated with culture medium for 2 h before transferring ≈3000 cells in suspension into each microwell.

**Inhibitor**. To inhibit actomyosin contractility we used Y27632, the cell permeable inhibitor of Rho kinase (ROCK), purchased from Merck Millipore, dissolved in water to make 10 mM stock solutions and used in final concentrations ranging from 50 to 100 µM. For negative control treatments, an identical amount of water was used.

**S100A4 constructs**. To inhibit the filament formation and function of non-muscle myosin 2 A in A431 epithelial carcinoma cells we transfected them with human S100A4 constructs. DNA constructs expressing either the wild type or an inactive mutant were prepared as previously described[34]. The mutant form (MutS100A4), derived using the Megaprimer method[48], lacks 13 amino acids at the C-terminal and contains a point mutation in position 81 that replaces a cysteine by serine, therefore does not inhibit NM2 assembly. Both the wild type and mutant S100A4 coding sequences were subcloned into pIRES2-eGFP plasmid (Clontech), which contains an internal ribosome entry site using restriction sites XhoI and BamHI. Using the BsaI restriction site for linearization, cells were transfected with linearized plasmids, using FuGene HD transfection reagent (Promega) according to the manufacturer's instructions. Stable transfectants were selected with 0.4 mg/ml G418 antibiotics (Merck Millipore). After two weeks of selection in G418-supplemented medium, stably transfected cells were further selected by their GFP signal using FACSAria Cell Sorter (BD Biosciences). After selection, cells were maintained in 0.2 mg/ml G418. Six S100A4-overexpressing and five mutS100A4-overexpressing cell clones were eventually established, of which one from each group was used for aggregation and segregation studies.

**Western blot**. A431 stable transfectants were lysed in cell lysis buffer (25 mM Tris pH 7.4, 150 mM NaCl, 2 mM EDTA, 2.5 mM DTT, 1% v/v Triton X-100, and 1% v/v protease inhibitor cocktail). Protein concentration was measured by Bradford assay, and 20 μg total protein samples were run on SDS-PAGE using 10% Tris-Tricine gel. Samples were blotted to PVDF membrane, then S100A4 protein was detected by anti-S100A4 antibody (mouse, monoclonal, PR006.21.3, 1:1000 dilution, a kind gift of Dr. Jörg Klingelhöfer of University of Copenhagen) followed by horseradish peroxidase-conjugated anti-mouse secondary antibody (1:5000 dilution, Santa Cruz). For loading control, tubulin was detected by anti-tubulin monoclonal antibody, (1:5000 dilution, Sigma) and horseradish peroxidase-conjugated anti-mouse antibody. Chemiluminescence was detected by using ECL Western Blotting Substrate (Pierce).

**Dual pipette aspiration assay (DPA)**. DPA was performed as described in detail earlier[23]. In summary, single cell suspensions were prepared from monolayer cultures of keratinocytes after brief incubation in PBS then washing off in $CO_2$-independent Medium (GIBCO) and seeded on passivated glass bottom dish (Mattek). Cells were collected using micropipettes prepared from glass capillaries (World Precision Instrument, TW100-3) using a micro-needle puller (Sutter Instrument, P-97 Flaming Brown micropipette puller) to produce capillaries with taper of approximately 3.5 μm internal radius, bent to 45° angle (Microdata instrument, MFG-5) and passivated with heat inactivated FBS (Invitrogen).

The micropipettes were connected on two independent channels with negative pressure ranging from 7–750 Pa to a Microfluidic Flow Control System (Fluigent, Fluiwell), mounted on two micromanipulators (Eppendorf, Transferman Nk2) with micropipette movement and pressure controlled via a custom-programmed Labview (National Instruments) interface. For cell manipulations approx. 20 Pa negative pressure was used.

After bringing the cells into contact, doublets were left to adhere for 1 min. The adhering cells were then grabbed by two micropipettes (holding and probing pipettes) on opposite sides of the cell-cell contact. The holding micropipette was used to firmly hold one cell with a fixed pressure ranging from 10–5000 Pa. The probing micropipette was used to apply stepwise increasing pressures ranging from 2–10 Pa with step sizes between 0.5 and 3 Pa to the other cell. After each pressure step, the micropipettes were pulled apart at 20 μm/s in an attempt to separate the contacting cells. Once the applied pressure in the probing pipette was high enough to separate the contacting cells, separation force, $F_s$, could be calculated from the final pressure and the pressure of the last failed separation using $F_s = pi R^2(P_n − 1 + P_n)/2$ where R is the micropipette radius, $P_n$ is the pressure applied by the probing pipette during the separating step and $P_n − 1$ is the pressure applied during the pulling step preceding the separating step. The number of pressure steps was usually kept between 1 and 3.

Cell separations were imaged on a Zeiss Axiovert inverted microscope equipped with an LD PlanNe- oFluar 40x, 0.6 NA Ph2 Korr objective, a CoolSnap HQ camera (Photometrics) and a heating box set to 28 °C. Cell doublet contact angles were characterized using NIH ImageJ software.

**Microscopy**. Time-lapse recordings with phase contrast or double fluorescence applications were performed on a Zeiss Axio Observer Z1 inverted microscope with Zeiss Plan Neofluar 10x, 0.3 NA objective coupled to a Zeiss Axiocam MRM CCD camera and equipped with a Marzhauser SCAN-IM powered stage. Zeiss Colibri LED illumination system was used for fluorescent excitation. For structured illumination microscopy and fluorescent optical sectioning we used Zeiss Apotome module. During time-lapse imaging the cell cultures were kept in a stage-top incubator (CellMovie) providing for required temperature and $CO_2$ atmosphere. Power stage positioning, illumination, focusing, optical sectioning and primary image collection were controlled by Zeiss Axiovision 4.8 software and a custom-made experiment manager software module on a PC. Images were further processed using Zeiss Axiovision 4.8 and NIH ImageJ software.

**Image analysis**. Images recorded by phase-contrast time-lapse microscopy were analyzed using the NIH ImageJ software with plugins of our design, available at https://github.com/gulyasmarton/ContractilityAnalyzer.git. For the spheroid aggregation assay, the images were preprocessed by an edge detection Sobel operator and a 5 pixels wide Gaussian blur filter. The preprocessed images were segmented by the default ImageJ thresholder (IsoData[49]). Segmentation noise was reduced by filling holes smaller than 1000 pixels. The spheroid was identified as the largest cluster of connected pixels. Repeating this procedure for each frame of the image sequence, we calculated the time-dependent perimeter of the spheroid.

**Cluster size analysis**. To characterize the average size of clusters in fluorescence microscopic images, correlation length analysis was performed as described earlier[47]. The correlation length of two-point correlations was used. Only the pixels with intensities above a threshold were considered.

First, we determined the two-point correlations for both colors in a given range (0–200 μm) and calculated the diameter of the clusters. For small distances, the two-point correlation can be approximated as a quadratic form of the distance:

$$C(r<R_0) \approx K(3/2R_0^2 − 2R_0r + 1/2r^2) \qquad (1)$$

where $R_0$ denotes the average diameter of clusters and K is a factor that is usually unknown as it
depends on the distribution of the cluster shapes. By fitting a polynomial to the correlations we obtained the typical size of a cluster.

**Statistics and reproducibility**. Where statistical analyses were performed, the relevant sample sizes as the number of independent entities analyzed are indicated as the n numbers. Where statistically significant differences are stated as the result of comparing the mean values of two data sets, Student's t-test was applied with an unpaired setting and assuming unequal variances. Generally, the p value disclosed refers to the two-tail P $(T <= t)$ probability.

For graphs showing the time-dependent cluster size growth of a particular cell type during segregation from another cell type, the data collection and analysis processes are the following: Image series of z-stacks of 2D optical sections encompassing the entire cell cluster volume are individually subjected to two-point correlation length analysis, yielding linear cluster size data on the basis of analysis of n > 1000 points in a single image. This analysis is repeated in all images of the z-stack of the time series originally recorded at 10 min intervals for various total durations ranging from 10 to 50 h, yielding 60 to 150 consecutive data points, specified for the given experimental condition. The number of cell clusters (i.e. individual 3D cell cultures) analyzed this way as independent entities are represented by the n number assigned to the specific experimental condition (cell type, treatment) and included in the legend of the relevant figure graph. These figure graphs show mean values +/− standard error of the mean (SEM) values calculated from cluster size data averaged from n > 1000 data points measured in 6 to10 z-planes for n number of independent cell cultures specified, and these are shown as individual data points in data series containing a time series 60 to 150 data points, depending on the specific experiment. For details see Supplementary Data 2 and Supplementary Data 4.

In the case of graphs where the included data are not time-dependent, such as separation force data for cell doublets (Fig. 1) or contact angle data for cell doublets (Fig. 7), the individual data points were transformed into box and whisker plots indicating the minimum, the lower quartile, the median, the upper quartile and the maximum values on the basis of the five-number summary set of descriptive statistics. The relevant sample numbers are included as the n number of the relevant data source, and in all cases, these are indicated in the related figure legends. For details see Supplementary Data 1 and Supplementary Data 5.

When time-dependent size data are included in figure graphs depicting the multicellular aggregation process of various cell types, such as Figs. 4 and 8, the following data collection and analysis method was used: Time-lapse image series of individual 3D cell cultures were collected at 10 min intervals for durations ranging from 10 to 25 h yielding raw data series of 60 to 150 data points for each independent sample, represented as the n number included in the relevant figure legend. For each data series, we set the data sampling to 1 h encompassing 6 time points and calculated the mean size value +/− standard error of the mean (SEM) values and plotted these values in the relevant graphs. For details see Supplementary Data 3.

Access to the numerical source data for all graphs is provided in the form of datasheets in the Supplementary Data files enclosed with this article.

**Reporting summary**. Further information on research design is available in the Nature Portfolio Reporting Summary linked to this article.

## Data availability

All data supporting the findings of this study are included in this published article and its Supplementary Information and Supplementary Data files.

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

## Acknowledgements

We thank Marton Gulyas (ELTE Eötvös University) for development of videomicroscopy experiment manager and image analysis software. Authors are grateful to Gabor Forgacs (University of Missouri) for critical reading of earlier versions of this manuscript as well as to Zsuzsa Akos and Andras Czirok (ELTE Eötvös University) for fruitful discussions. This work was supported by EU FP7, ERC COLLMOT Project No 227878 to TV, the National Research Development and Innovation Fund of Hungary, K119359 and also Project No 2018-1.2.1-NKP-2018-00005 to LN. This project has received funding from the European Union's Horizon 2020 research and innovation programme under the Marie Sklodowska-Curie grant agreement No 955576. MV was supported by the Ja´nos Bolyai Fellowship of the Hungarian Academy of Sciences.

## Author contributions

E.M. and T.V. conceived the project and designed the experiments, wrote the manuscript; E.M. performed the experiments with support from M.V., A.Z. and G.K. for zebrafish experiments, B.B.-K. and L.N. for S100A4 genetic manipulations; V.B. and E.Méhes performed DPA experiments; E.M. performed data analysis with support from E.Mones for cluster size analysis; C.-P.H. contributed to interpretation of study data. All authors reviewed the manuscript.

## Funding

## Competing interests

The authors declare no competing interests.
