## [Peer Review File · Communications Biology]

Reviewers' comments:

Reviewer #1 (Remarks to the Author):

Sorting and spatial segregation of cells with different biomechanical properties play an important role in tissue organization during embryonic development. Various hypotheses have been proposed to explain the spatial pattern of cell segregation on the bases of cell adhesion and cortical tension. Non-muscle myosin II (NMII) contractility has been implicated in cell segregation due to its function in generating cortical tension and influencing cell-cell adhesion. In this manuscript, Mehes et al investigated the impact of manipulating NMII contractility on the dynamics and spatial pattern of cell segregation using three pairs of model cell types, fish primary keratinocytes (PFK) and keratinocyte cell line (EPC), human epithelial carcinoma (A431) and fibrosarcoma cells (HT1080), and zebrafish ectoderm and mesoderm cells. The authors found that pharmacological inhibition of Rho kinase (ROCK), an activator of NMII, reduced the rate of cell segregation without affecting the inside-outside relationship of the segregated domains. Next, the authors show that inhibition of NMII assembly by expression of S100A4 in A431 cells resulted in reduced rate of cell aggregation from homotypic suspension and inversed segregation when mixed with HT1080 cells. Finally, the authors show that enhancing NMII activity in zebrafish ectoderm cells by expression of constitutively active ROCK (caROCK) resulted in reduced cell contact angle, moderately impaired homotypic cell aggregation and inverted sorting behavior in progenitor cell mixtures.

The study addresses an important question in the mechanics of tissue organization. The experiments and data analyses are carefully performed and are of high quality. The manipulation of cortical contractility in various cell types along with the quantitative analysis of segregation dynamics provide useful data for understanding the mechanics of cell sorting.

My main questions lie in the interpretation of the data, in particular the proposed impacts of manipulating NMII activity on cortical tension and cell adhesion in connection with the observed cell segregation phenotypes. The detailed comments are listed below.

Major comments:

1. Since the role of NMII in regulating tissue surface tension and cell sorting has been previously demonstrated in the process of zebrafish progenitor segregation, the authors should more clearly explain the motivation of the present study and the novelty of the findings.

2. In figure 1 and 2, the authors measured adhesive strength of the PFK and EPC cells and characterized their sorting behavior. While the outcome is consistent with the prediction of differential adhesion hypothesis, the potential contribution of differential cortical tension between the two cell types should also be discussed. In addition, regarding the segregation pattern of zebrafish ectoderm and mesoderm cells, the authors state that "...and eventually ectoderm cells segregated to the inside while mesoderm cells to the outside (Fig. 2c,

Supplementary Movie 5), in line with expectations based on earlier data showing higher homotypic contact strengths for ectoderm cells compared to mesoderm cells [23. 32].” However, in a different study, it has been shown that ectoderm cells are the least cohesive cell type (PMID: 18364700).

3. NMII contractility has been shown to be important for the formation of cell-cell adhesions in certain epithelia (e.g. PMID: 20543839). In the experiments presented in Figure 3, could the defect in cell segregation following ROCK inhibitor treatment be due to impaired cell-cell adhesion instead of reduced cortical tension? Similarly, was cell adhesion affected by expression of s100A4?

4. It is puzzling why both decreasing NMII activity (Figure 4 ,5) and increasing NMII activity (Figure 6-8) resulted in similar inverted positioning of segregated domains. Based on the mechanism elaborated in this manuscript, the segregation of cell types is determined by differential tissue surface tension (TST), which is defined as:

$TST = \text{cell-medium interface tension (Tcm)} - \text{cell-cell interface tension (Tcc)}$

whereas $Tcm = \text{total cortical tension at the cell-medium interface.}$

and $Tcc = \text{total cortical tension at the cell-cell interface} - \text{total cell adhesion tension.}$

In theory, inhibiting NMII contractility should cause reduction in both Tcm and Tcc . Conversely, enhancing NMII contractility should cause increase in both Tcm and Tcc . However, in both cases, the cell segregation outcomes suggest a reduction in TST. The authors should clarify the impact on Tcm and Tcc for each NMII manipulation. In addition, the level of NMII at cell-medium and cell-cell interfaces should be examined and compared between the control group, the s100A4-expression group and the caROCK-expression group, which could both confirm the effect of the treatments and serve as a proxy for cortical tension at specific interfaces.

5. The effects of caROCK expression in contact angle (Figure 6) and aggregation dynamics (Figure 7) are quite mild. Given the striking difference in contact angle between ectoderm and mesoderm cells as previously reported (PMID: 22923438), it is unclear whether the changes caused by caROCK expression is sufficient to revert the difference between the ectoderm and mesoderm cells. The conclusion of this set of experiments would be further strengthened if the authors could directly measure and compare TST between ectoderm-caROCK and mesoderm cell aggregates, but I understand that this experiment may be beyond the scope of the current study.

Minor points:

1. Figure 1b,c, the contact angle should be quantified.

2. For the dual pipette aspiration assay, “stepwise increasing pressures ranging from 10-4000 Pa with step sizes between 10 and 500 Pa” was applied. It seems that the range of step size is rather large. The authors should explain how the step size is actually chosen in the experiments.

3. The final spatial organization of the segregated cells was not always clear in the images/movies presented (e.g. Fig 3d and 3f). It is unclear in these cases how the inside-outside relationship is determined.
4. The mechanism underlying the observed segregation pattern for the A431 and HT1080 cell types should be introduced (differential cell adhesion, differential cortical tension, or both?).
5. In some cases, statistical test is missing from the figure/figure legend (e.g., Figure 1a, Figure 4a, Figure 7a; And plots for aggregation dynamics if applicable).
6. For the sentence “For comparison, we overexpressed an inactive truncated mutant isoform of S100A4 (MutS100A4), eliciting no effect on NM2 assembly, in another A431 subclone (termed A431-ctrl) used as negative control[33]”, citation 33 should be 35.

Reviewer #2 (Remarks to the Author):

In this manuscript, Méhes et al use various cell pairs to study how different perturbations of actomyosin-generated contractility influence the dynamics of cell segregation in a three-dimensional culture model. From their results, the authors conclude that general inhibition of actomyosin contractility delays segregation, while leaving the spatial configuration of the segregated cell types unaffected. They further manipulate Myosin activity in specific cell types, either through inhibition of Myosin IIA assembly by overexpression of S100A4 (specifically in A431 cells that are cocultured with HT1060 cells) or through increasing myosin contractility by overexpression of a constitutively active mutant of ROCK (specifically in zebrafish ectoderm cells that are cocultured with mesoderm cells). Their findings show that S100A4 overexpression alters the dynamics of segregation of A431-HT1060 cells, and results in an inverted cell distribution with A431 cells now segregating on the outside instead of inside of the spheroid cultures. This inverted geometry in segregation is similarly observed upon CA-ROCK expression in zebrafish ectoderm cells, which also causes defective cell-cell contact formation and aggregation of ectoderm cells (the latter is similarly shown for S100A4 overexpression in A431 cells).

Although the data in this manuscript is clearly presented and in general well analyzed, my main concern is that I find it difficult to understand how this work extends on previous work that is also cited by the authors in the introduction. In particular, Krieg et al 2008 has shown how ectoderm-mesoderm segregation is affected by selective actomyosin inhibition in ectoderm progenitor cells, i.e. with ectoderm cells sorting to the outside instead of inside of heterotypic aggregates with endoderm or mesoderm cells. Both the current manuscript and Krieg et al use a comparable assay, and although the analysis in the current manuscript is extended by segregation

dynamics it is not clear to me how the conclusions are different.

In contrast to previous work, the authors also include experiments in which actomyosin contractility is increased through CA-ROCK expression. However, in my opinion the authors do not provide a clear explanation why they observe similar effects with CA-ROCK as with S100A4 overexpression. If the authors aim to claim that this is due to interface-specific regulation of cortical tension (with CA-ROCK expression preventing local downregulation of actomyosin contractility at the cell-cell interface) this should be directly tested by manipulating the connection between cell-cell adhesions and myosin regulation (e.g. by using p120-catenin mutants that attenuate its ability to regulate myosin, as described in the references cited by the authors).

Furthermore, the experiments with S100A4 and CA-ROCK overexpression are performed in different cell types. For direct comparison of these results, it is important to perform both types of manipulation in at least one of the used cell lines.

The authors conclude that Y27632 treatment does not affect the final spatial configuration of the heterotypic cultures, with cell types normally segregating inside still taking the inner positions under ROCK inhibition. This is not very clear from the provided images and movies. For instance, the proposed segregation of A431 to the inside of spheroids is not very apparent in Fig. 2d; and it is difficult to see how this differs from A431-S100A4 expressing cells in Fig. 4d in which case it is concluded they these cells segregate outside the spheroids. It is therefore important to support these data with analysis of the cellular distributions.

The comparison of A431 spheroid size and morphology in Fig. 4 should also include cells lacking expression of either of the constructs, because from the current data it can not be concluded whether there is an effect of the wildtype S100A4 protein or instead of expression of the mutant.

Reviewers' comments:

We thank both Reviewers for their encouraging and thoughtful remarks as well as for their helpful suggestions. Addressing their insightful comments has contributed significantly to an improvement of the original version of our manuscript. We reproduce all of their specific points below (in italics) together with our responses. Parts of our responses which appear in the new manuscript are with colored (blue) font. Superscripts follow the numbering of references listed in the submitted manuscript.

Inspired by Reviewers' comments we decided to slightly modify even the title of our manuscript for "The complex geometry and dynamics of three-dimensional cell segregation are governed by the regulation of tissue surface tension in multiple cell types" to indicate both main messages of our work.

Reviewer #1 (Remarks to the Author):

After summarizing the main features of our work, Reviewer #1 notes that our *"... study addresses an important question in the mechanics of tissue organization. The experiments and data analyses are carefully performed and are of high quality. The manipulation of cortical contractility in various cell types along with the quantitative analysis of segregation dynamics provide useful data for understanding the mechanics of cell sorting."*

We thank Reviewer #1 for finding the topic of our manuscript important and of high quality. Below we address the points raised in the report one by one.

Major comments:

1. Since the role of NMII in regulating tissue surface tension and cell sorting has been previously demonstrated in the process of zebrafish progenitor segregation, the authors should more clearly explain the motivation of the present study and the novelty of the findings.

The segregation of cells is a phenomenon during which both highly specific biological and less specific physical factors play important roles. The emerging spatial structures may or may not depend on details of the underlying biology depending on the degree of the more universal physical features of the process, e.g., on such quantities as the level of surface tension or adhesion. In addition to the morphology, the time scales over which a particular type of pattern develops is of essential relevance because it is likely to be closely related to the time scale over which an embryo develops.

Therefore, we added to the Discussion part of our manuscript:

The main mechanisms acting while a fertilized egg develops into a fully developed embryo is still not completely understood, but some of the ingredients of the process can be identified as being dominated by diffusion, spontaneous collective migration and migration as a result of some guiding signals (e.g. gradients of chemokines and/or growth factors)¹⁸.

In the light of the above considerations we were motivated to investigate two aspects of the emergence of morphology during 3D cell segregation: i) we tested whether the conclusions of prior studies about the pattern formation of segregating zebrafish progenitor cells apply to a wider range of vertebrate cells and ii) we applied a computational analysis to the time lapse 3D recordings, which opens the way for analyzing the relative roles of the diffusion and group motion mechanisms during segregation.

The two modalities – diffusion and collective motion – result in distinct characteristic time dependencies. During the segregation of two types of cells of about the same number, theory predicts that the characteristic linear size of homotypic cell clusters grows with time as power law according to t^z with $z= 1/3$, while the coherent motion of cells of the same type results in a much faster pace of segregation characterized by a linear growth of the cluster diameters with $z= 1$, accordingly³⁹. Since one of the focuses of our study is the time dependence of the level of segregation, it was essential to analyze our measured cell configurations by a well-established method borrowed from statistical physics and relying on determining the so-called density correlation function. Our quantitative data on the dynamics of various in vitro models of 3D cell segregation can be used by future studies on computational modeling of 3D segregation.

As regards the novelty of our work concerning the “*role of NMII in regulating tissue surface tension and cell sorting*”, we were motivated by earlier studies on zebrafish progenitor segregation and adhesion, and we also extended these studies and added substantial novel insights.

The study by Krieg et al., 2008 (pmid 18364700) showed the impact of both a pharmacological inhibitor of NMII and a dominant negative ROCK isoform on progenitor segregation where the readout was the final geometry of segregated domains. In our study we extend observations of similar perturbations of NMII activity into the time-resolved quantitative analysis of the dynamics of segregation. Here we provide new analysis tools developed from well-established statistical physical methods, which can measure characteristic cell cluster size (1D, characteristic length) data from 3D reconstruction images in a reliable way independent of cluster shape.

Therefore we completed the revised Introduction with these lines:

Motivated by earlier studies on the role of actomyosin cytoskeleton in the regulation of tissue surface tension and segregation^{21,23}, we extended these studies to a wider range of vertebrate cell types and developed a quantitative analysis of the dynamics of segregation.

In their seminal work Maitre et al., 2012 (pmid 22923438) proved that effective cell-cell contact formation in germ layer progenitor cell doublets requires active reduction of actomyosin-based cortical tensions at the contacting cell-cell interfaces. We extended their findings into

experiments with large number of cells, combined with the analysis of segregation dynamics. We have proved that at multicellular level, the generation of tissue surface tension depends on cells' ability to differentially regulate their cortical tensions in an interface-specific manner. Using constitutively active ROCK isoform as an experimental tool we also showed that this interface-specific regulation is based on the activity of ROCK, the main NMII activator, which is downregulated by signaling from the cell-cell contact interface. Thus we provided experimental support on the molecular mechanism enabling cells to make contacts and aggregate or segregate.

Accordingly, we rephrased the relevant parts of Discussion (quoted in response to major comment #4) and included the following:

The above findings, when compared to earlier studies^{21,23}, jointly provide experimental proof that interface-specific tuning of cortical tension through the local regulation of the actomyosin contractile system by ROCK determines TST of multicellular structures, which drives their self-organization, including segregation.

2. In figure 1 and 2, the authors measured adhesive strength of the PFK and EPC cells and characterized their sorting behavior. While the outcome is consistent with the prediction of differential adhesion hypothesis, the potential contribution of differential cortical tension between the two cell types should also be discussed. In addition, regarding the segregation pattern of zebrafish ectoderm and mesoderm cells, the authors state that "...and eventually ectoderm cells segregated to the inside while mesoderm cells to the outside (Fig. 2c, Supplementary Movie 5), in line with expectations based on earlier data showing higher homotypic contact strengths for ectoderm cells compared to mesoderm cells [23. 32]." However, in a different study, it has been shown that ectoderm cells are the least cohesive cell type (PMID: 18364700).

Response to comment #2:

We fully agree with the Reviewer that in view of the different adhesive strengths measured for PFK and EPC keratinocytes (Fig.1) their sorting behavior (Fig.2) is consistent with the differential adhesion hypothesis. However, these experiments indeed do not allow us to assess the contribution of differential cortical tension because the separation forces - measured by the dual pipette aspiration assay - also depend on cortical tension regulation in the cell pairs. The potential contribution of differential cortical tension here can be deduced from segregation experiments with pharmacological inhibition of NMII activity in both cell types (Fig.3a,b), where it is observed that such inhibition (using high inhibitor concentrations) has little effect on final sorting geometry while slowing the dynamics of segregation. Therefore it seems that in this cell pair system the contribution of differential cortical tension to the generation of overall TST difference may be secondary.

In line with the Reviewer's request, we have included a short discussion of this phenomenon in the relevant part of Discussion:

Quantitative analysis of the dynamics of segregation can reveal here the sensitivity of the dynamics to general inhibition of actomyosin contractility and this can also reveal the different contributions of cortical tensions and adhesion tension to TST generation.

The rate of cluster size growth in keratinocyte segregation is little affected under actomyosin inhibition (Fig. 3a), indicating higher contribution of adhesion to TST generation, as compared to the segregation of A431 and HT1080 cells where inhibition of actomyosin contractility profoundly slows down segregation (Fig. 3c), indicating higher contribution of cortical tensions to TST.

As regards the contradiction between the contact strength of ectoderm reported by Maitre et al., 2012 (referred to by our manuscript) and Krieg et al., 2008 (also referred to by the Reviewer) the key lies in the duration of observations using these two methods. Krieg et al., 2008 reported contact strengths measured by single cell force spectroscopy 10 to 60 seconds after contact formation yielding $F_s=3$ nN. On the other hand, Maitre et al. reported contact strengths measured by dual pipette aspiration assay 60 to 600 seconds after establishment of contact, while contact strength of ectoderm was steadily on the increase from $F_s=13$ nM up to $F_s=18$ nN, while mesoderm remained below $F_s=10$ nN. The duration of the latter measurement is more relevant for our long term observations and their findings in contact strength are in harmony with the segregation geometry we observed.

3. NMII contractility has been shown to be important for the formation of cell-cell adhesions in certain epithelia (e.g. PMID: 20543839). In the experiments presented in Figure 3, could the defect in cell segregation following ROCK inhibitor treatment be due to impaired cell-cell adhesion instead of reduced cortical tension? Similarly, was cell adhesion affected by expression of s100A4?

Response to comment #3:

We agree with the Reviewer that we cannot rule out that inhibiting ROCK can indeed affect cell adhesion as NMIIA regulated by ROCK was reported to act on adhesion receptor clustering (reference cited by the Reviewer).

However, quantitative analysis of segregation dynamics provides a new tool to assess the contribution of adhesion tension and cortical tensions to TST generation.

The data presented in Fig.3/a,b argue against high impact of ROCK inhibition on adhesion because even with decreased cortical tensions the segregation of PFK and EPC epithelial cells is just delayed without any change in final geometry, so their respective TST generation here can be dominated by the unaffected differences between their respective adhesion tensions.

We elaborated the interpretation of this finding in Discussion where we now write:

This may be due to the fact that in lack of considerable NM2-based cortical tensions the unperturbed differences in adhesion tension can now dominate TST generation and drive slower segregation. An alternative interpretation is that the applied concentrations of the ROCK inhibitor, although falling in the high range for the studied cell types³⁴, can allow cell type-

specifically different residual NM2 activities accounting for different cortical tensions resulting in the unchanged segregation configurations.

For comparison, the normal segregation of A431 epithelial cells and HT1080 fibroblasts seems to be dominated by differences in their respective cortical tensions because ROCK inhibition here severely slows down the dynamics of segregation (Fig3c,d). Based on the latter observation we can assume that expression of S100A4 in A431 cells decreases TST and slows down segregation mainly by decreasing cortical tensions (Fig. 5c).

We added a summary of these two different contribution patterns in Discussion:

The rate of cluster size growth in keratinocyte segregation is little affected under actomyosin inhibition (Fig. 3a), indicating higher contribution of adhesion to TST generation, as compared to the segregation of A431 and HT1080 cells where inhibition of actomyosin contractility profoundly slows down segregation (Fig. 3c), indicating higher contribution of cortical tensions to TST.

4. It is puzzling why both decreasing NMII activity (Figure 4 ,5) and increasing NMII activity (Figure 6-8) resulted in similar inverted positioning of segregated domains. Based on the mechanism elaborated in this manuscript, the segregation of cell types is determined by differential tissue surface tension (TST), which is defined as:

TST = cell-medium interface tension (Tcm) – cell-cell interface tension (Tcc)

whereas Tcm = total cortical tension at the cell-medium interface.

and Tcc = total cortical tension at the cell-cell interface – total cell adhesion tension.

In theory, inhibiting NMII contractility should cause reduction in both Tcm and Tcc. Conversely, enhancing NMII contractility should cause increase in both Tcm and Tcc. However, in both cases, the cell segregation outcomes suggest a reduction in TST. The authors should clarify the impact on Tcm and Tcc for each NMII manipulation. In addition, the level of NMII at cell-medium and cell-cell interfaces should be examined and compared between the control group, the s100A4-expression group and the caROCK-expression group, which could both confirm the effect of the treatments and serve as a proxy for cortical tension at specific interfaces.

Response to comment #4:

We thank the Reviewer for calling our attention to an important discrepancy in phrasing the observations and interpretations regarding caROCK manipulations (submitted version Figures 6-8). Our emphasis should be on caROCK's effect on: i) increasing the cell-cell interface tension (Tcc) by not letting actomyosin contractility be locally downregulated by endogenous ROCK, which is normally downregulated via trans-bound cadherin -> catenin -> ROCK negative regulation pathway, while ii) caROCK also increases cortical tension at cell-medium interface (Tcm) but to a lesser extent because actomyosin contractility here is already high owing to the normal activity of endogenous ROCK, due to lack of local downregulation in absence of trans-bound cadherins here.

In summary, for the caROCK case: lower than normal TST = moderately increased high Tcm - highly increased Tcc, compared to the normal case when TST = normal high Tcm –

downregulated low Tcc, this is our interpretation based on the shift in contact angle and differences in aggregation dynamics.

We agree with Reviewer that we have to be cautious with the interpretation of the impact of NMII manipulation on Tcm and Tcc and therefore we carefully rephrased all relevant parts of the manuscript to clarify these ambiguities. Additionally, we have included a new combined schematic figure summarizing: i) what we currently know about the generation of tissue surface tension on the basis of published works, and ii) providing an overview of our experimental interventions aiming to verify the role of ROCK in interface-specific regulation of cortical tension.

We have also elaborated the corresponding explanatory part in Discussion:

All these observations indicate that caROCK, which cannot be downregulated, constantly activates NM2 and thus keeps actomyosin contractility high at all sites of the cell cortex, while cortical tension should normally be reduced at the cell-cell interface due to inactivation of endogenous ROCK by signaling from trans-bound cadherins here. Conversely, due to lack of trans-bound cadherins at the cell-medium interface, endogenous ROCK is not inactivated there by this signaling pathway and keeps actomyosin contractility high, which is not assumed to be considerably further increased by caROCK. Because TST primarily depends on the difference of cortical tension at the cell-medium interface (Tcm) and cortical tension at the cell-cell interface (Tcc) with minor contribution of cell-cell adhesion tension (Acc) as $TST \approx T_{cm} - T_{cc} + Acc/2$ in general^{2, 3, 23, 41}, this difference is diminished in caROCK-expressing cells due to their decreased ability to reduce cortical tension at the cell-cell interface, while the minor contribution of cell-cell adhesion tension is not assumed to change (Fig. 6).

And finally, we also agree with the Reviewer that measuring and comparing NMII levels at the cell-medium vs. cell-cell interfaces would certainly support the efficacy of our NMII manipulations, however such experiments are currently not feasible in the timeframe for this major revision.

5. The effects of caROCK expression in contact angle (Figure 6) and aggregation dynamics (Figure 7) are quite mild. Given the striking difference in contact angle between ectoderm and mesoderm cells as previously reported (PMID: 22923438), it is unclear whether the changes caused by caROCK expression is sufficient to revert the difference between the ectoderm and mesoderm cells. The conclusion of this set of experiments would be further strengthened if the authors could directly measure and compare TST between ectoderm-caROCK and mesoderm cell aggregates, but I understand that this experiment may be beyond the scope of the current study.

Response to comment #5:

We understand Reviewer's concerns regarding the mild (but statistically significant) effects of caROCK on contact angle (decrease) and aggregation dynamics (delayed and decreased compaction). However, the important findings of these two independent tests are that both of them indicate a tendency of decrease in tissue surface tension when interface-specific cortical tension regulation is perturbed by caROCK expression. Regardless of the extent of change in

these measurable parameters, the unequivocal negative tendency of change in both parameters is informative of the role of ROCK as an interface-specific regulator of cortical tension.

Accordingly, we rephrased the relevant part of Discussion:

Slightly delayed and reduced compaction during multicellular aggregation of caROCK-expressing ectoderm cells is also characteristic along with the appearance of loosely adhering round cells at the surface of these spheroids, indicating a tendency of reduced TST (Fig. 8). Importantly, when homotypic ectoderm cell doublets are formed in culture by cell-cell adhesion, we observe a tendency of reduced contact angles in caROCK-expressing cell doublets (Fig. 7), which is an indication of increased cell-cell interfacial tension here.

As an alternative of detecting changes in TST on the basis of cell doublet contact angles or aggregation dynamics, we turned to the sorting assay as a multicellular direct evaluation ‘tool’ where cells themselves compare the aggregate tissue surface tensions of the respective cell types and spatially organize in accordance with their respective cell type-specific TST level. Inverted sorting behavior of mesoderm and caROCK-ectoderm with characteristic changes in the dynamics was observed in several independent experiments, enabling us to conclude that the TST of caROCK ectoderm not only decreased but even became lower than the TST of (untreated) mesoderm.

To elaborate this interpretation, we compiled a new schematic figure summarizing the expected effects of caROCK expression, and included these lines in Results, chapter **2.6 Constitutively active ROCK causes defective contact formation and aggregation**:

For making it easier to interpret our experiments we compiled a schematic figure summarizing: i) current knowledge about cytoskeleton regulation by cell adhesion molecules, ii) the major physical components generating tissue surface tension as it is currently known^{2, 3, 23}, and iii) the expected consequences of our experimental interventions into cell contractility regulation (Fig. 6).

Figure 6. Schematic summary of experimentally studied aspects of tissue surface tension regulation. a) Signaling from cell adhesion molecules to the cytoskeleton. Trans-binding of cell surface cadherins of contacting cells results in decreased actomyosin contractility and cortical tension at the cell-cell interface (T_{cc}). Green arrows indicate activation, red symbol represents inhibition. Detailed description of this signaling cascade and references are included in Discussion. b) Summary of tissue surface tension (TST) generation by cells in an aggregate, shown as usual schematic representation. Cell cortical tension at the cell-medium interface (T_{cm}) and adhesion tension (A_{cc}) contribute positively to TST whereas cortical tension at the cell-cell interface (T_{cc}) has negative contribution. Tensions are represented by arrows. A contact angle Θ is indicated by dotted lines as a guide to the eye. c-d) Schematic images of cell aggregates with key components of cortical tension regulation highlighted by symbols to explain the experimental interventions. c) Normal cells at the surface of the aggregate showing effective multicellular compaction characterized with large contact angle Θ due to relaxed actomyosin network at the cell-cell interface and low cortical tension as a result of signaling from trans-bound cadherins. d) Genetically manipulated cells expressing constitutively active ROCK isoform, which constantly activates actomyosin contractility at the cell-cell interface regardless of inactive endogenous ROCK here. Eventually, caROCK maintains higher cortical tension at the cell-cell interface, leading to reduced TST and less effective compaction characterized with smaller contact angle. Definitions of symbols are shown in central text box.

Finally, we added an explanatory part to Discussion (same as referred to in our response to major comment #4).

We nevertheless fully agree with the Reviewer that other direct measurement of TST between ectoderm-caROCK and mesoderm cell aggregates would further support our interpretation but such experiments would clearly not be feasible in this study, at least not in the timeframe available for major revision.

Although the Reviewer does not pose it as being a major concern, we would like to call attention to the fact that in our assay the untreated ectoderm doublets yield lower contact angles than similar ectoderm reported by Maitre et al., 2012. The main difference between the two assays is that ectoderm contact angles were monitored for a duration up to 10 minutes by Maitre et al., 2012 (being on the increase while mesoderm doublet contact angles were on the decrease), whereas we measured ectoderm contact angles after 30 minutes (with no comparable data on mesoderm doublets).

Minor points:

1. Figure 1b,c, the contact angle should be quantified.

Response to minor #1:

We have quantified the contact angles in Fig. 1b,c and included the data in the figure image.

2. For the dual pipette aspiration assay, “stepwise increasing pressures ranging from 10-4000 Pa with step sizes between 10 and 500 Pa” was applied. It seems that the range of step size is rather large. The authors should explain how the step size is actually chosen in the experiments.

Response to minor #2:

We have checked the minutes recorded for the DPA assays, and corrected these data in Methods. The range of the step size was 0.5 – 3 Pa and the pressure range was 2 – 10 Pa.

3. The final spatial organization of the segregated cells was not always clear in the images/movies presented (e.g. Fig 3d and 3f). It is unclear in these cases how the inside-outside relationship is determined.

Response to minor #3:

The inside-outside relationship at the endpoint in these segregation experiments (Fig.3d, Fig.3f) were determined qualitatively “by the eye” on the basis of the images, without any quantitative measures for the spatial organization of the homotypic domains. We fully understand Reviewer’s concern and to remove potential ambiguity concerning the final spatial organizations we have included a new supplementary figure (panel) showing 2 additional examples for the final spatial organization for each cell type pair and each treatment type, all from independent experiments. These supplementary figures are included in our response to comment #4 of Reviewer #2. These supplementary figures include images showing some morphological variations (concerning size, shape) and hopefully help the reader capture the similarities regarding the inside-outside relationship. Additionally, we have updated Fig.3d and Fig.3f by including other examples (images) where the spatial organization can be better captured by the eye. We have also updated the corresponding supplementary movies (Movie7 and Movie8) to show more easily traceable examples of delayed segregation with unchanged spatial configurations.

4. The mechanism underlying the observed segregation pattern for the A431 and HT1080 cell types should be introduced (differential cell adhesion, differential cortical tension, or both?).

Response to minor #4:

The mechanism underlying the segregation of A431 and HT1080 cells is the interplay of differential cell adhesion and differential cortical tension. Yet we think the major contribution is provided by differential cortical tension, as supported by experiments with ROCK inhibition (Fig.3c,d) where segregation of A431 / HT1080 became very slow in lack of normal actomyosin activity. The mechanism underlying these findings are now discussed in the relevant part of Discussion and also quoted in our response to major comment #2 above.

5. In some cases, statistical test is missing from the figure/figure legend (e.g., Figure 1a, Figure 4a, Figure 7a; And plots for aggregation dynamics if applicable).

Response to minor #5:

We have included the missing statistical test data in the figure legends (Figure 1a, Figure 4a, submitted version Figure 7a, which is Figure 8a in the revised version).

6. For the sentence “For comparison, we overexpressed an inactive truncated mutant isoform of S100A4 (MutS100A4), eliciting no effect on NM2 assembly, in another A431 subclone (termed A431-ctrl) used as negative control[33]”, citation 33 should be 35.

Response to minor #6:

Here, we would like to adhere to the reference of our choice because reference No 33 in the first submission version is actually the study (Mehes et al., 2019, co-authored by several authors of this manuscript) where the A431 subclones overexpressing the S100A4 isoforms (wt or inactive mutant) and their contractile functions are described in detail. Therefore this study is referenced for the negative control “A431-ctrl” clone in our manuscript.

Reviewer #2 (Remarks to the Author):

Reviewer #2 starts his review with an insightful summary of the results presented in our manuscript and starts his comments by noting that “...the data in this manuscript is clearly presented and in general well analyzed...”. We thank Reviewer #2 for their assessment indicating the quality features of our manuscript.

1) Although the data in this manuscript is clearly presented and in general well analyzed, my main concern is that I find it difficult to understand how this work extends on previous work that is also cited by the authors in the introduction. In particular, Krieg et al 2008 has shown how ectoderm-mesoderm segregation is affected by selective actomyosin inhibition in ectoderm progenitors cells, i.e. with ectoderm cells sorting to the outside instead of inside of heterotypic aggregates with endoderm or mesoderm cells. Both the current manuscript and Krieg et al use a comparable assay, and although the analysis in the current manuscript is extended by segregation dynamics it is not clear to me how the conclusions are different.

Response to #1 comment

Our work is aimed to extend the findings of previous works into the field of multicellular segregation of cell types of various origins, along with quantitative analysis of segregation dynamics. Previous works include those studying the role of actomyosin-based contractility in segregation (Krieg et al., 2008, pmid 18364700) and also those revealing the basic mechanism of cell-cell contact formation (Maitre et al., 2012, pmid 22923438).

On the basis of these earlier findings our study aims to prove that interface-specific regulation of NM2 activity and the resultant differential cortical tension has instructive role not only in cell contact formation but also in tissue surface tension (TST) generation and therefore in 3D aggregation and segregation of multiple cells. To this end we went further in manipulating NM2 regulation using the well-established zebrafish germ layer progenitor segregation system,

similarly to Krieg et al., but also including time-resolved quantitative analysis of the dynamics of 3D segregation. In their study Krieg et al. used dominant negative ROCK construct to ubiquitously inhibit actomyosin contractility in ectoderm cells and demonstrated a resultant decrease in TST. In our study we expressed a constitutively active ROCK isoform to interfere with cortical tension regulation of ectoderm cells and found decreased contact formation and TST.

To interpret our findings we expanded Discussion (also quoted in response to major comment #3 of Reviewer #1):

All these observations indicate that caROCK, which cannot be downregulated, constantly activates NM2 and thus keeps actomyosin contractility high at all sites of the cell cortex, while cortical tension should normally be reduced at the cell-cell interface due to inactivation of endogenous ROCK by signaling from trans-bound cadherins here. Conversely, due to lack of trans-bound cadherins at the cell-medium interface, endogenous ROCK is not inactivated there by this signaling pathway and keeps actomyosin contractility high, which is not assumed to be considerably further increased by caROCK. Because TST primarily depends on the difference of cortical tension at the cell-medium interface (T_{cm}) and cortical tension at the cell-cell interface (T_{cc}) with minor contribution of cell-cell adhesion tension (Acc) as $TST \approx T_{cm} - T_{cc} + Acc/2$ in general^{2, 3, 23, 41}, this difference is diminished in caROCK-expressing cells due to their decreased ability to reduce cortical tension at the cell-cell interface, while the minor contribution of cell-cell adhesion tension is not assumed to change (Fig. 6).

Additionally, we have included a new combined schematic figure summarizing: i) what we currently know about TST generation, and ii) providing an overview of our experimental interventions aiming to verify the role of ROCK in interface-specific regulation of cortical tension.

This figure is included in response to major comment #5 of Reviewer #1

2) In contrast to previous work, the authors also include experiments in which actomyosin contractility is increased through CA-ROCK expression. However, in my opinion the authors do not provide a clear explanation why they observe similar effects with CA-ROCK as with S100A4 overexpression. If the authors aim to claim that this is due to interface-specific regulation of cortical tension (with CA-ROCK expression preventing local downregulation of actomyosin contractility at the cell-cell interface) this should be directly tested by manipulating the connection between cell-cell adhesions and myosin regulation (e.g. by using p120-catenin mutants that attenuate its ability to regulate myosin, as described in the references cited by the authors).

Response to #2 comment)

We understand the Reviewer's concern about the necessity of a clear explanation of caROCK's effects. We attempt to give one below as well as in the revised Discussion of our manuscript.

The manipulations to decrease actomyosin contractility either by S100A4 overexpression or by dominant negative ROCK expression (in the study of Krieg et al., 2008) are not interface-specific but act ubiquitously in the cell and eventually reduce TST by decreasing the normal

difference between cortical tensions at the cell-medium and the cell-cell interfaces. Conversely, we interpret the results of caROCK expression as an important support of the existence of an interface-specific regulation of cortical tension.

We included an explanation of this in Discussion, which is quoted in our response to #1 comment.

The regulatory axis from trans-bound cadherins via catenins to inhibition of ROCK is established in literature although the parts of this axis were characterized in different experimental systems. For a focused summary on the molecular mechanism of interface-specific cortical tension regulation we have included a new schematic figure (quoted in response to comment #5 of Reviewer #1) and added the following to the relevant part in Discussion:

The implied mechanism involves p120-catenin activation following its recruitment to cadherin complexes²⁸. Activated p120-catenin inhibits RhoA activity locally²⁷ through inhibition of p190RhoGAP³⁰ leading to further downstream inactivation of ROCK and eventually NM2 activity. Alpha-catenin contributes to the binding and activation of p120-catenin⁴⁰ and the importance of this interaction is indicated by studies showing that loss of alpha-catenin in embryonic cell aggregates results in reduced tissue surface tension⁴¹. The signaling cascade starting from trans-bound cadherins eventually downregulates NM2-based actomyosin contractility, enabling local control of cortical tension (Fig. 6).

Because the main regulator of NM2 activity is ROCK and it is the first upstream regulator of NM2 in the cadherin-catenin-RhoA-ROCK regulatory axis (although other signaling pathways may also act on ROCK at the cell-cell interface) we specifically manipulated ROCK, as a minimal model. However, caROCK is not a cell-cell interface-specific tool. Overexpression of caROCK caused not only prevention of local downregulation of actomyosin contractility at the cell-cell interface (normally done by endogenous ROCK) but also must have increased NM2 activity at cell-medium interface, nevertheless this increase is thought to be minor as endogenous ROCK is already fully active there due to lack of local downregulation in absence of trans-bound cadherins. Major increase in actomyosin contractility at the cell-cell interface and minor increase at the cell-medium interface eventually decreases the difference of these two cortical tensions and results in reduction of overall tissue surface tension. Therefore although caROCK is not an interface-specific tool for manipulating cortical tension, yet its effects are more pronounced at the cell-cell interface and this way it reveals the role of normal ROCK in the interface-specific regulation of cortical tension.

We agree with the Reviewer that an interface-specific tool for manipulating cortical tension (i.e. cadherin or catenin mutants) would further support the existence of interface-specific cortical tension regulation but such manipulations fall outside the scope of this study and would not be feasible in the timeframe available for major revision.

3) Furthermore, the experiments with S100A4 and CA-ROCK overexpression are performed in different cell types. For direct comparison of these results, it is important to perform both types of manipulation in at least one of the used cell lines.

Response to #3 comment)

We agree with the Reviewer that it is useful to perform various manipulations in the same experimental system in order to e.g. decrease the variability of experimental results due to variability of the experimental systems. Our caROCK manipulations in zebrafish progenitors can be compared with the study by Krieg et al., 2008 that used dnROCK in the same cell type to inactivate NM2. Yet, we also intended to show that manipulations of cortical tensions have comparable effects on cell aggregation/segregation in quite distant experimental systems, suggesting universality in the mechanism. Therefore we performed our S100A4 manipulations (to inactivate NM2) in differentiated human cells, implying the universal nature of these phenomena. In order to call attention to this comparison, we added the following to the relevant part of Discussion:

Similar reduction in TST was demonstrated earlier²¹ upon inactivation of NM2 by the expression of a dominant negative ROCK isoform in zebrafish ectoderm cells, suggesting a universal mechanism in cell types of various origins.

We additionally note that performing S100A4 and caROCK manipulations in a single experimental system is currently not feasible for us for two reasons:

- i) S100A4 is absent in zebrafish where the closest homologue is S100Z (with as little as 47% homology, Moroz et al., pmid: 21756915), so we would probably not have any effect with our human S100A4 construct.
- ii) The caROCK construct (used for RNA microinjection in zebrafish zygote) is expected to be hard to introduce in human A431 cell line as a stably expressed construct such as S100A4.

4) The authors conclude that Y27632 treatment does not affect the final spatial configuration of the heterotypic cultures, with cell types normally segregating inside still taking the inner positions under ROCK inhibition. This is not very clear from the provided images and movies. For instance, the proposed segregation of A431 to the inside of spheroids is not very apparent in Fig. 2d; and it is difficult to see how this differs from A431-S100A4 expressing cells in Fig. 4d in which case it is concluded they these cells segregate outside the spheroids. It is therefore important to support these data with analysis of the cellular distributions.

Response to #4 comment)

Having carefully reviewed all images and videos we agree with Reviewer that the final segregation positioning or (in case of A431 / HT1080 segregation) the tendency of positioning is difficult to see in some cases.

While development of an analysis of cellular distributions to support the current (visual) evaluation of segregation positioning is not feasible in the timeframe of this major revision, we improved and expanded the image data supporting our findings.

Therefore, we introduced the following changes into the revised version:

In order to provide better examples where segregation positioning can more easily be captured by the eye we collected additional representative images of the final segregation positioning from independent experiments and included 3 new supplementary figures linked to parts of Results summarizing: **1) ROCK inhibition, 2) S100A4 manipulation and 3) caROCK manipulation experiments.** In these supplementary figures two additional independent examples

of final segregation positioning are shown for each cell type pair and each manipulation. We think these extra images provide parallel examples where it is easier to capture the segregation positioning tendencies in spite of the morphological variations of the individual cases.

1) In the revised manuscript, we made the following change in Results, part 2.3
Dynamics of segregation depends on actomyosin contractility:

The cell types normally segregating inside still tended to take the inner positions under ROCK inhibition while the fusion of emerging inner clusters into a single central cluster generally did not occur within the normal timeframe (Fig. 3b,d,f, **Supplementary Figure 1**, Supplementary Movies 6-8).

We included the following figure in the Supplementary Information:

Supplementary Figure 1

3D segregation is delayed by general pharmacological inhibition of actomyosin contractility.

a) Representative 3D reconstruction images from time-lapse videos of segregating clusters of PFK (red) and EPC (green) keratinocytes 6 h after start of the segregation process in absence (left panels) or presence (right panels) of 100 μM Y27632 ROCK inhibitor.

b) Images of A431 epithelial carcinoma (red) and HT1080 fibrosarcoma (green) segregation in absence (left) or presence (right) of 50 μM Y27632, after 24 h segregation. Note the formation and delayed fusion of multiple clusters of A431 cells inside the spheroids under ROCK inhibition (right).

c) Representative images of ectoderm (red) and mesoderm (green) segregation in absence (left) or presence (right) of 100 μM Y27632, after 15 h segregation.

Time elapsed after initial mixing of heterotypic cell suspensions is indicated at the lower right corner of each image, scale bars: 100 μm . The vertical yellow line indicates the border between left and right panels.

2) We made the following change in Results, part 2.5 **Selective inhibition of non-muscle myosin 2 assembly leads to inverted segregation:**

This inverted spatial positioning of homotypic domains indicates that stable inhibition of actomyosin contractility in A431-S100A4 cells eventually reduced their specific tissue surface tension below that of HT1080 cells (**Supplementary Figure 4**).

We included the following figure in the Supplementary Information:

Supplementary Figure 4

Spatial positioning during segregation depends on non-muscle myosin 2 assembly and function.

a) Representative 3D reconstruction images from time-lapse videos of HT1080 fibrosarcoma cells (red) segregating from A431-ctrl epithelial carcinoma cells (green) overexpressing the

inactive mutant S100A4 isoform. Note the segregation of A431-ctrl (green) clusters to the inside of spheroids.

b) Representative 3D images of segregation of HT1080 cells (red) from A431-S100A4 cells (green) overexpressing the NM2 assembly inhibitor S100A4. Note the tendency of A431-S100A4 (green) clusters to segregate to the periphery of spheroids.

Time elapsed after initial mixing of heterotypic cell suspensions is indicated at the lower right corner of each image, scale bars: 100 μm .

3) Finally, we made the following change in Results, part 2.7 **Cell type specific constitutively active ROCK causes inverted segregation:**

Mesoderm cells now tended to segregate to the inside of spheroids with normal cell cluster growth dynamics, while ectoderm-caROCK cluster growth rate was steadily reduced (Fig. 9c), and ectoderm-caROCK clusters segregated to the outside as an indication of decreased tissue surface tension (**Supplementary Figure 5**).

We included the following figure in the Supplementary Information:

Supplementary Figure 5

Spatial configuration of segregated domains depends on actomyosin contractility regulation.

a) Representative 3D reconstruction images from time-lapse videos of mesoderm cells (green) segregating from normal ectoderm cells (red) isolated from zebrafish embryos. Note the segregation of ectoderm (red) clusters to the inside of spheroids.

b) Representative 3D images of segregation of mesoderm cells (green) from ectoderm-caROCK cells (red) overexpressing constitutively active ROCK. Note the tendency of ectoderm-caROCK (red) clusters to segregate to the periphery of spheroids.

Time elapsed after initial mixing of heterotypic cell suspensions is indicated at the lower right corner of each image, scale bars: 100 μ m.

***[end of legend]

Additionally, we have updated the videos showing the segregation of A431/HT1080 cells (Movie7) or ectoderm/mesoderm cells (Movie8) under ROCK inhibition and included examples where the phenomenon is easier to see.

We have also updated Figure 3 (ROCK inhibition) and included representative images in panel d and f where segregation positioning under ROCK inhibition is more apparent. (We respectfully note that the Reviewer must have aimed their concerns to Figure 3d as Fig.2d does not exist and similarly it is Figure 5d where A431-S100A4-expressing cells' inverted segregation is shown while Fig.4d, pointed to by the Reviewer, does not exist.)

5) The comparison of A431 spheroid size and morphology in Fig. 4 should also include cells lacking expression of either of the constructs, because from the current data it cannot be concluded whether there is an effect of the wild type S100A4 protein or instead of expression of the mutant.

Response to #5 comment)

In full agreement with Reviewer's notice we have completed and included a new experiment for comparison of the size and morphology of normal A431 spheroids lacking any S100A4 (wt or inactive mutant) constructs. As a summary, we have added a new supplementary figure (linked to Figure 4) where representative images of spheroids of a) normal A431 cells, b) inactive mutant S100A4-expressing A431-ctrl cells, and c) wt S100A4-expressing A431-S100A4 cells are compared at the final phase of aggregation (24 h after onset of aggregation with equal cell number). Normal A431 and A431-ctrl spheroids resemble each other concerning their final compacted size and also their smooth surface (indicating high tissue surface tension) and they characteristically differ from A431-wtS100A4 spheroids, which are generally larger and less compact and have berry-like surface (indicating lower TST). This additional 'control' experiment supports our conclusion that the TST reducing effect in A431 cells is indeed due to the effect of wild type S100A4 overexpression.

Accordingly, we made the following change in Results, part 2.4 **Inhibition of non-muscle myosin 2 assembly drives defective aggregation:**

We placed suspensions of equal number of cells from each A431 subclone separately in non-adherent aggregation chamber wells and observed formation of spheroid aggregates by time-lapse videomicroscopy (Supplementary Movie 9, **Supplementary Figure 3**).

We also included the following figure in the Supplementary Information:

Supplementary Figure 3

Aggregation morphology depends on non-muscle myosin 2 assembly and function.

a) Representative phase-contrast image from a time-lapse video of normal A431 cells forming a spheroid after 24 h of aggregation from suspension.

b) Image of a spheroid formed of A431 cells overexpressing the inactive mutant isoform of S100A4 (A431-ctrl). Note the smooth surface of the compact spheroids in a) and b) panels.

c) Representative image of a spheroid formed of A431 cells overexpressing the NM2 assembly inhibitor S100A4 (A431-S100A4). Note the berry-like surface and less compact morphology of the spheroid in c).

Time elapsed after start of aggregation from cell suspension is indicated at the lower right corner of each panel image, scale bar: 100 μm.

Reviewers' comments:

Reviewer #1 (Remarks to the Author):

The authors have addressed most of my previous questions in their revised manuscript. The expanded discussion and the schematic figure (Fig. 6) greatly clarify the proposed explanation of the experimental observations. I believe the manuscript is suitable for publication after some minor modifications:

(1) Fig. 6 is a very helpful addition. Since the purpose of this figure is to demonstrate “the expected consequences of our experimental interventions into cell contractility regulation”, I suggest adding a cartoon panel to show the expected impact of overexpressing S100A4 (similar to Fig. 6d) and placing Fig. 6 before Fig. 4.

(2) In the figure legend for Fig. 6d, the authors should indicate that the schematic demonstrates the PROPOSED impact of expressing CA-ROCK on cortical actomyosin at the cell-medium interface and the cell-cell interface, since the actual impact still awaits experimental validation (e.g. by examining the level of NMII at cell-medium and cell-cell interfaces).

(3) Regarding Fig. 1b, 1c, the authors conclude that “the contact shapes were also different as PFK doublets were observed to be characterized by higher contact angles than EPC doublets”. This conclusion should be backed up with proper quantifications (like in new Fig. 7b). Currently only the angles of one pair of cells were shown for each cell type. (Also see my previous Minor point 1).

(4) In their response to my previous Major point 2, the authors stated that “these experiments indeed do not allow us to assess the contribution of differential cortical tension because the separation forces - measured by the dual pipette aspiration assay - also depend on cortical tension regulation in the cell pairs”. It would be helpful to include a brief explanation in the discussion how cortical tension regulation impacts separation forces.

(5) Some images in Supplementary figure 1 are duplicates of the images in Fig 3. These images should be replaced with new examples.

-Supplementary figure 1b ctrl right panel is identical to Fig. 3d ctrl

-Supplementary figure 1b Y27632 right panel is identical to Fig. 3d Y27632

-Supplementary figure 1c ctrl right panel is identical to Fig. 3f ctrl

-Supplementary figure 1c Y27632 left panel is identical to Fig. 3f Y27632

Reviewer #2 (Remarks to the Author):

In their revised manuscript, Méhes et al. have addressed most of my initial concerns, particularly

by the inclusion of additional analyses of cellular distributions and by clarification of the motivation of the study and interpretation of the results.

However, one of my main concerns has not been resolved. For the interpretation of the data, the authors fully rely on the assumption that ca-ROCK overexpression induces interface-specific regulation of cortical tension. Although I appreciate the detailed discussion about this from the authors in the rebuttal letter, without any experimental evidence I am not convinced that effects of ca-ROCK are more pronounced at the cell-cell interface than the cell-medium interface. The authors for instance state that the increase at the cell-medium interface is thought to be minor as endogenous ROCK is already fully active here: but even though there will not be cadherin-induced downregulation of ROCK activity that does not exclude that only a subfraction of ROCK is active at the cell-medium interface. If more specific interface-specific manipulations (suggested in my initial comments) are out of the scope of the current manuscript, then at the very least the authors should have examined and compared the level of NM2 (total levels and/or phosphorylation status of MLC) at the cell-medium and cell-cell interfaces under the different conditions, as also indicated by the other Referee. Without this key experimental evidence, I am not convinced that the conclusions are sufficiently supported by the presented data.

Furthermore (as explained in more detail in my original comments), to compare the results of experiments with S100A4 and ca-ROCK overexpression I think it is essential to perform both of these manipulations in the same cell type. I fail to see why it is not feasible to express ca-ROCK in mammalian cell lines (i.e. A431s). I appreciate it may be difficult to generate clones stably expressing ca-ROCK, but I don't see why either transient expression and/or inducible expression would not have been possible.

Minor comment:

There appear to some mistakes in the explanation about NM2 regulation downstream of E-cadherin in the added paragraphs (e.g. page 7). Inhibition of p190RhoGAP would lead to increased RhoA activity (instead it is the recruitment of p190RhoGAP that leads to reduced RhoA activity). P120-catenin is not an enzyme and thus does not have "activity".

Reviewers' comments in report No 2:

We thank both Reviewers for their new remarks as well as for their helpful suggestions. We reproduce all of their specific points below (in italics) together with our responses. Parts of our responses which appear in the new manuscript are with colored (blue) font. Superscripts follow the numbering of references listed in the submitted manuscript.

Reviewers' comments:

Reviewer #1 (Remarks to the Author):

The authors have addressed most of my previous questions in their revised manuscript. The expanded discussion and the schematic figure (Fig. 6) greatly clarify the proposed explanation of the experimental observations. I believe the manuscript is suitable for publication after some minor modifications:

We thank the Reviewer for these supportive and encouraging remarks.

(1) Fig. 6 is a very helpful addition. Since the purpose of this figure is to demonstrate “the expected consequences of our experimental interventions into cell contractility regulation”, I suggest adding a cartoon panel to show the expected impact of overexpressing S100A4 (similar to Fig. 6d) and placing Fig. 6 before Fig. 4.

Response to comment (1):

We have created a cartoon panel showing the expected impact of S100A4 overexpression and added this panel to Fig. 4 originally depicting the aggregation dynamics with representative microscopic images of spheroids of the control vs. S100A4 overexpression groups.

Figure 4. Aggregation dynamics depends on non-muscle myosin 2 assembly. Quantitative analysis of aggregation of A431 cells. a) Time-dependent decrease in the mean perimeter of spheroids aggregating from homotypic cell suspension of A431 cells overexpressing either NM2 assembly inhibitor S100A4 (A431-S100A4, n=10) or its inactive mutant isoform (A431-ctrl, n=10). Error stripes represent SEM, asterisk (*) at t=24 h indicates statistically significant difference with Student's t-test, $p < 0.05$. b) Representative phase-contrast images from time-lapse videos of spheroids of A431-ctrl cells (left panel) or A431-S100A4 cells (right panel) after 24 h aggregation from suspension. Note the difference in spheroid surface roughness. Scale bar: 100 μm . Also see Supplementary Movie 9 and Supplementary Figure 3. c-d) Schematic representations of surface cells highlighting the cytoskeletal components involved in cortical tension generation. c) Normal cells with effective multicellular compaction characterized by large contact angle Θ due to the contracted actomyosin network at the cell-medium surface and relaxed actomyosin at the cell-cell interface coupled by cadherins. Assembly of NM2 monomers into filaments is controlled by normal levels of S100A4, assembly and disassembly processes are symbolized by the double-headed arrow. d) Experimentally increased levels of S100A4 leads to sequestration of NM2 monomers and shifting towards the disassembly of filaments, assumed to result in a shift towards reduced cortical actomyosin tension and reduced multicellular compaction. Definitions of symbols are shown in central text box.

(2) *In the figure legend for Fig. 6d, the authors should indicate that the schematic demonstrates the PROPOSED impact of expressing CA-ROCK on cortical actomyosin at the cell-medium interface and the cell-cell interface, since the actual impact still awaits experimental validation (e.g. by examining the level of NMII at cell-medium and cell-cell interfaces).*

Response to comment (2):

We have rephrased the legend for Fig. 6d in accordance with Reviewer's comment. Now it is rephrased as:

“The proposed impact is that caROCK maintains higher cortical tension at the cell-cell interface, leading to reduced TST and less effective compaction characterized by smaller contact angle.”

Additionally, we have performed a set of new experiments on the impact of caROCK on aggregation dynamics and eventually examined the level of active NM2 in the aggregated cells by immunofluorescent detection of the phosphorylated myosin regulatory light chain. These new results are incorporated in Fig. 8 originally showing the aggregation dynamics of ectoderm cells and representative images of the spheroids from the control vs. caROCK overexpression groups.

(3) *Regarding Fig. 1b, 1c, the authors conclude that “the contact shapes were also different as PFK doublets were observed to be characterized by higher contact angles than EPC doublets”. This conclusion should be backed up with proper quantifications (like in new Fig. 7b). Currently only the angles of one pair of cells were shown for each cell type. (Also see my previous Minor point 1).*

Response to comment (3):

We have measured all contact angles recorded in the relevant set of DPA experiments and included the average and SEM data corresponding to these contact angles in the main text where Fig. 1 is referred to.

Contact shapes were also different as PFK doublets were observed to have larger contact angles (45.1°, SEM=0.71) than EPC doublets (35.5°, SEM=0.65), in harmony with the differences in contact strength.

(4) In their response to my previous Major point 2, the authors stated that “these experiments indeed do not allow us to assess the contribution of differential cortical tension because the separation forces - measured by the dual pipette aspiration assay - also depend on cortical tension regulation in the cell pairs”. It would be helpful to include a brief explanation in the discussion how cortical tension regulation impacts separation forces.

Response to comment (4):

We have included a brief explanation in Discussion how cortical tension regulation impacts separation forces.

The contact strength of two contacting cells can be characterized by the separation force, which is experimentally measurable *ex vivo* by the dual pipette aspiration (DPA) assay. Besides the separation force, the contact angle characterizing the contacting cells is also indicative of the contact strength as higher contact angles coincide with higher separation forces. Contact strength is dependent on cell surface adhesion molecules, contributing positively, and also limited by their anchorage to the cytoskeleton, while a negative contribution comes from the tension of the cell cortex at the site of contact. Cortical tension at the contact is actively reduced during contact formation of germ layer progenitor cells, running along with a reduction of NM2 at the contact area²³.

(5) Some images in Supplementary figure 1 are duplicates of the images in Fig 3. These images should be replaced with new examples.

-Supplementary figure 1b ctrl right panel is identical to Fig. 3d ctrl

-Supplementary figure 1b Y27632 right panel is identical to Fig. 3d Y27632

-Supplementary figure 1c ctrl right panel is identical to Fig. 3f ctrl

-Supplementary figure 1c Y27632 left panel is identical to Fig. 3f Y27632

Response to comment (5):

In Supplementary Figure 1 referred to by the Reviewer we have replaced these images with new examples from independent sets of experiments. Originally, the duplicates of Fig. 3 images were in this Supplementary Figure on purpose to show their similarity with independent examples, but we agree with Reviewer that additional independent examples further increase comparability.

Reviewer #2 (Remarks to the Author):

In their revised manuscript, Méhes et al. have addressed most of my initial concerns, particularly by the inclusion of additional analyses of cellular distributions and by clarification of the motivation of the study and interpretation of the results.

We thank the Reviewer for these supportive and encouraging remarks.

(1) However, one of my main concerns has not been resolved. For the interpretation of the data, the authors fully rely on the assumption that ca-ROCK overexpression induces interface-specific regulation of cortical tension. Although I appreciate the detailed discussion about this from the authors in the rebuttal letter, without any experimental evidence I am not convinced that effects of ca-ROCK are more pronounced at the cell-cell interface than the cell-medium interface. The authors for instance state that the increase at the cell-medium interface is thought to be minor as endogenous ROCK is already fully active here: but even though there will not be cadherin-induced downregulation of ROCK activity that does not exclude that only a subfraction of ROCK is active at the cell-medium interface. If more specific interface-specific manipulations (suggested in my initial comments) are out of the scope of the current manuscript, then at the very least the authors should have examined and compared the level of NM2 (total levels and/or phosphorylation status of MLC) at the cell-medium and cell-cell interfaces under the different conditions, as also indicated by the other Referee. Without this key experimental evidence, I am not convinced that the conclusions are sufficiently supported by the presented data.

Response to comment 1):

We have performed a new set of experiments on the effect of caROCK overexpression on the aggregation of ectoderm cells. As suggested by the Reviewer we have chosen to examine the activity of NM2 under the different conditions (ctrl vs. caROCK overexpression) as immunofluorescence detection of the phosphorylated myosin regulatory light chain (p-MLC). In Fig. 8 originally showing the aggregation dynamics, we have included comparable representative image panels of p-MLC immunolabeling in spheroids from the two groups. Using identical labeling and imaging settings we have found that compared to ectoderm-ctrl aggregates the pMLC levels are considerably higher throughout ectoderm-caROCK aggregates with no apparent difference between surface and bulk.

Accordingly, we have rephrased the relevant part of Discussion where we now write:

Conversely, due to lack of trans-bound cadherins at the cell-medium interface, endogenous ROCK is not inactivated there by this signaling pathway and keeps actomyosin contractility high, which can be further increased by caROCK. Because TST primarily depends on the difference of cortical tension at the cell-medium interface (T_{cm}) and cortical tension at the cell-cell interface (T_{cc}) with minor contribution of cell-cell adhesion tension (Acc) as $TST \approx T_{cm} - T_{cc} + Acc/2$ in general^{2,3,23,42}, this difference is diminished in caROCK-expressing cells due to their decreased ability to reduce cortical tension at the cell-cell interface, while the minor contribution of cell-cell adhesion tension is not assumed to change (Fig. 6).

We have also performed quantitative analysis of the aggregation dynamics of these new experiments and added the results to the data of former similar experiments and we have

accordingly updated the aggregation dynamics graph in Fig. 8a as well as the corresponding statistical analysis data.

Figure 8. Aggregation dynamics is influenced by regulation of actomyosin contractility. Quantitative analysis of aggregation of zebrafish ectoderm cells. a) Time-dependent decrease in the mean perimeter of spheroids aggregating from homotypic cell suspensions of either untreated ectoderm cells (ectoderm-ctrl, n=20) or ectoderm cells overexpressing constitutively active ROCK (ectoderm-caROCK, n=20). Spheroid perimeters are normalized with and plotted as percentage of initial perimeters, error stripes represent SEM, asterisk (*) at t=10 h indicates statistically significant difference with Student's t-test, $p < 0.05$. b) Representative phase-contrast images from time-lapse videos of spheroids of ectoderm-ctrl cells (left panel) or ectoderm-caROCK cells (right panel) after 10 h of aggregation. Note the difference in spheroid surface roughness. Scale bar: 100 μ m. Also see Supplementary Movie 12. c-f) Immunofluorescent detection of phospho-myosin light chain (p-MLC) in spheroids of ectoderm cells. c-d) Ectoderm-ctrl spheroid with only p-MLC labeling (c) or merged image of p-MLC (yellow) and NucBlue (blue) labels. e-f) A spheroid of ectoderm-caROCK cells labeled for p-MLC (e) or merged image of p-MLC label (yellow) and NucBlue (blue) staining. Cell nuclei are visualized by NucBlue staining. Note that the p-MLC immunofluorescence signal is hardly seen in (c) while it is much more pronounced after the introduction of caROCK in (e). Scale bar: 10 μ m in c-f).

(2) Furthermore (as explained in more detail in my original comments), to compare the results of experiments with S100A4 and ca-ROCK overexpression I think it is essential to

perform both of these manipulations in the same cell type. I fail to see why it is not feasible to express ca-ROCK in mammalian cell lines (i.e. A431s). I appreciate it may be difficult to generate clones stably expressing ca-ROCK, but I don't see why either transient expression and/or inducible expression would not have been possible.

Response to comment 2):

The main reason for expressing caROCK in the zebrafish progenitor system is that by microinjecting a well-defined amount of caROCK mRNA at 2-cell stage we can control the amount of caROCK protein expressed during the following hours when the proliferation, dissociation, then aggregation or segregation of ectoderm cells take place. Even in this controlled system the fine tuning of the effective (high) amount of mRNA to yield the required phenotype took us several months of experimentation. This mRNA microinjection-based expression control is clearly something we are unable to do in a mammalian cell line like A431, while we believe that transient and/or inducible expression would not result in high enough levels of caROCK and therefore unlikely to yield a meaningful outcome.

(3) *Minor comment:*

There appear to some mistakes in the explanation about NM2 regulation downstream of E-cadherin in the added paragraphs (e.g. page 7). Inhibition of p190RhoGAP would lead to increased RhoA activity (instead it is the recruitment of p190RhoGAP that leads to reduced RhoA activity). P120-catenin is not an enzyme and thus does not have "activity".

Response to comment #3:

We have corrected erroneous phrasing in Discussion where we now write:

The implied mechanism involves p120-catenin recruitment to cadherin complexes²⁸, where p120-catenin inhibits RhoA activity locally²⁷ through the recruitment of p190RhoGAP³⁰. Inhibition of RhoA leads to further downstream inactivation of ROCK and eventually downregulation of NM2 activity.